# The chromatin remodeller ATRX facilitates diverse nuclear processes, in a stochastic manner, in both heterochromatin and euchromatin

Julia Truch[1], Damien J. Downes [1], Caroline Scott [1], E. Ravza Gür[1,2], Jelena M. Telenius [1,2], Emmanouela Repapi [2], Ron Schwessinger[1,2], Matthew Gosden [1], Jill M. Brown [1], Stephen Taylor [2], Pak Leng Cheong [1], Jim R. Hughes [1,2], Douglas R. Higgs [1✉] & Richard J. Gibbons [1✉]

The chromatin remodeller ATRX interacts with the histone chaperone DAXX to deposit the histone variant H3.3 at sites of nucleosome turnover. ATRX is known to bind repetitive, heterochromatic regions of the genome including telomeres, ribosomal DNA and pericentric repeats, many of which are putative G-quadruplex forming sequences (PQS). At these sites ATRX plays an ancillary role in a wide range of nuclear processes facilitating replication, chromatin modification and transcription. Here, using an improved protocol for chromatin immunoprecipitation, we show that ATRX also binds active regulatory elements in euchromatin. Mutations in ATRX lead to perturbation of gene expression associated with a reduction in chromatin accessibility, histone modification, transcription factor binding and deposition of H3.3 at the sequences to which it normally binds. In erythroid cells where downregulation of α-globin expression is a hallmark of ATR-X syndrome, perturbation of chromatin accessibility and gene expression occurs in only a subset of cells. The stochastic nature of this process suggests that ATRX acts as a general facilitator of cell specific transcriptional and epigenetic programmes, both in heterochromatin and euchromatin.

[1] MRC Molecular Haematology Unit, MRC Weatherall Institute of Molecular Medicine, Radcliffe Department of Medicine, University of Oxford, Oxford OX3 9DS, UK. [2] MRC WIMM Centre for Computational Biology, MRC Weatherall Institute of Molecular Medicine, Radcliffe Department of Medicine, University of Oxford, Oxford OX3 9DS, UK. ✉email: doug.higgs@imm.ox.ac.uk; richard.gibbons@ndcls.ox.ac.uk

The X-linked α-thalassaemia intellectual disability syndrome (ATRX) protein is a member of the SWI/SNF family of chromatin remodelling factors which acts as an ATP-dependent molecular motor[1]. Together with the histone chaperone DAXX, ATRX is involved in inserting the histone variant H3.3 at sites of nucleosome turnover in a DNA replication-independent manner. The importance of the ATRX/DAXX/H3.3 complex in processing chromatin throughout the genome is highlighted by the diverse nuclear activities that are perturbed when its components are mutated or knocked out. In particular, loss of ATRX affects DNA replication, DNA repair, homologous recombination, genome stability, DNA methylation, chromatin modification and gene expression[2–7]. Germline mutations in ATRX give rise to a severe form of syndromal intellectual disability (ATR-X syndrome) characteristically associated with downregulation of α-globin expression[8]. In addition, over the past 10 years, acquired mutations in ATRX have been very frequently found in tumours which maintain their telomeres via the alternative lengthening of telomere (ALT) pathway activated in 10–15% of all cancers[9]. Understanding the normal role of ATRX and the mechanisms by which mutations lead to inherited and acquired human disease is consequently of great current interest.

Most previous studies have shown that ATRX is located at a wide range of heterochromatic tandem repeats throughout the genome. These include rDNA, telomeric, pericentric and minisatellite repeats as well as endogenous retroviral sequences and the 3′ exons of the highly duplicated genes of the zinc finger family[3,10–14]. Common features of these repeats are that they are frequently transcribed, may form abnormal DNA structures, including G quadruplex (G4) DNA and most are noted as regions that are difficult to replicate. At these sites, ATRX appears to maintain the repressive chromatin marks that characterized such heterochromatic loci.

Previous efforts to determine any role of ATRX in euchromatin have been severely hampered by the quality of chromatin immunoprecipitation (ChIP) experiments. Here, using a much-improved protocol for ChIP-seq we have now comprehensively and definitively analysed the binding of ATRX within euchromatin in different cell types. We confirm that ATRX binds a large proportion of zinc finger genes and also a large subset of enhancers, promoters and gene bodies. At these transcriptionally active elements, there is known to be a high turnover of H3.3 during interphase[15] and we show here that in addition to its role at heterochromatin, the presence or absence of ATRX correlates with the levels of H3.3 at these regions of euchromatin which are normally bound by ATRX. Importantly, by comparing different cell types, we show that ATRX is recruited to regions defined by the *process* of transcription rather than by any specific subset of transcription factors. We also show that when ATRX is mutated, and where gene expression is perturbed, this is associated with changes in chromatin accessibility, histone modifications, binding of transcription factors and deposition of the histone variant H3.3 at the associated enhancers and/or promoters. Of interest, when analysing the role of ATRX in single cells, we show that in the absence of ATRX changes in chromatin structure at regulatory elements and consequent gene expression occur in a stochastic manner.

Detailed analysis of the role of ATRX at the well-characterized α-globin locus in erythroid cells illustrates how mutations in many widely expressed chromatin remodelling factors may lead to stochastic changes in gene expression and cell fate via a wide range of nuclear processes which they facilitate. In the case of ATRX, this may occur predominantly via interdependent roles in DNA replication, chromatin modification and chromatin accessibility.

## Results

**ATRX ChIP-seq confirms ATRX association with heterochromatin and reveals a variety of ATRX binding sites in euchromatin.** We performed ChIP-seq of endogenous ATRX, with two distinct antibodies, in human lymphoblastoid cell lines (LCLs) derived from three independent unaffected individuals (TA-Ctr, FF-Ctr and CB-Ctr). Using a significantly improved protocol[16] we produced sensitive and reproducible ATRX ChIP-seq datasets associated with low background inputs (Supplementary Fig. 1a). This identified 8,454 ATRX binding sites genome-wide and we selected a subset of ATRX-enriched regions to confirm our results by ChIP-qPCR (Supplementary Fig. 1b). We confirmed the affinity of ATRX for repetitive regions including pericentromeric DNA, rDNA and GC-rich, low complexity repeat sequences by direct mapping and sequence analysis at peak-called regions. In addition, we identified ATRX enrichment at several genes encoding tRNA and the small nuclear RNA U2 (Supplementary Fig. 2a and b). This analysis also showed ATRX associated with the KRAB domain-containing Cys2-His2 zinc fingers (C2-H2 ZNF) (Supplementary Fig. 2c) consistent with observations in other human cell types[14].

ATRX has been previously shown to bind directly to G4 in vitro and putative quadruplex sequence (PQS) containing targets in vivo[3] and here we observed that almost half of the ATRX binding sites in LCLs overlap PQS (Supplementary Fig. 2d). Peak calling revealed two main patterns of ATRX binding sites including broad shape peaks, similar to those observed with histone marks such as H3K9me3, contrasting with sharper peaks, usually observed with transcription factors (Fig. 1ai and aii) possibly reflecting different modes of ATRX recruitment.

With the reduced background afforded by the modified ChIP-seq protocol[16], it is now evident that, using short-read sequencing, most mappable ATRX targets are present at a subset of genes and many regions of increased chromatin accessibility. Almost two thirds of ATRX binding sites are intragenic with 25% spanning gene promoters, and 39% within gene bodies (Fig. 1b). ATRX binding sites were significantly enriched at transcription start sites (TSS) compared to matching size random fragments ($p$-value < 2.2e−16, odds ratio = 23.3, Fisher's exact test) (Supplementary Fig. 3a).

We further analysed the average distribution of ATRX across the entire genome and observed a strong signal enrichment at TSS (Fig. 1c). ATRX binding sites were enriched in annotated CpG islands which were present in 29% (compared to 2% in matching size random fragments, $p$-value < 2.2e−16, odds ratio = 18.9, Fisher's exact test) (Fig. 1d and Supplementary Fig. 3b). Using a BioCAP-seq approach, which detects non-methylated CpG islands[17], we found that most of the ATRX bound CpG islands are unmethylated and associated with promoter regions (Fig. 1d, e and Supplementary Fig. 4). At these sites, ATRX peaks were enriched in the active chromatin mark H3K4me3 and depleted of H3K9me3 signal (Fig. 1e and f). Only 113 ATRX binding sites were fully methylated on both alleles ($p$-value < 2.2e−16, odds ratio = 0.1, Fisher's Exact Test, compared to matching size random fragments) (Fig. 1d and Supplementary Fig. 3c); such sites were correlated with high levels of H3K9me3 (Fig. 1f) representative of heterochromatic regions to which ATRX is known to be recruited[18,19].

Given the strong association, we found here between ATRX and euchromatin, we investigated chromatin accessibility at ATRX binding sites. Using ATAC-seq we found that 87% of ATRX binding sites associate with regions of open chromatin, in line with ATRX enrichment at transcriptionally active chromatin regions (Fig. 1e, g). Comparison with matching size random fragments showed that ATRX was significantly enriched at

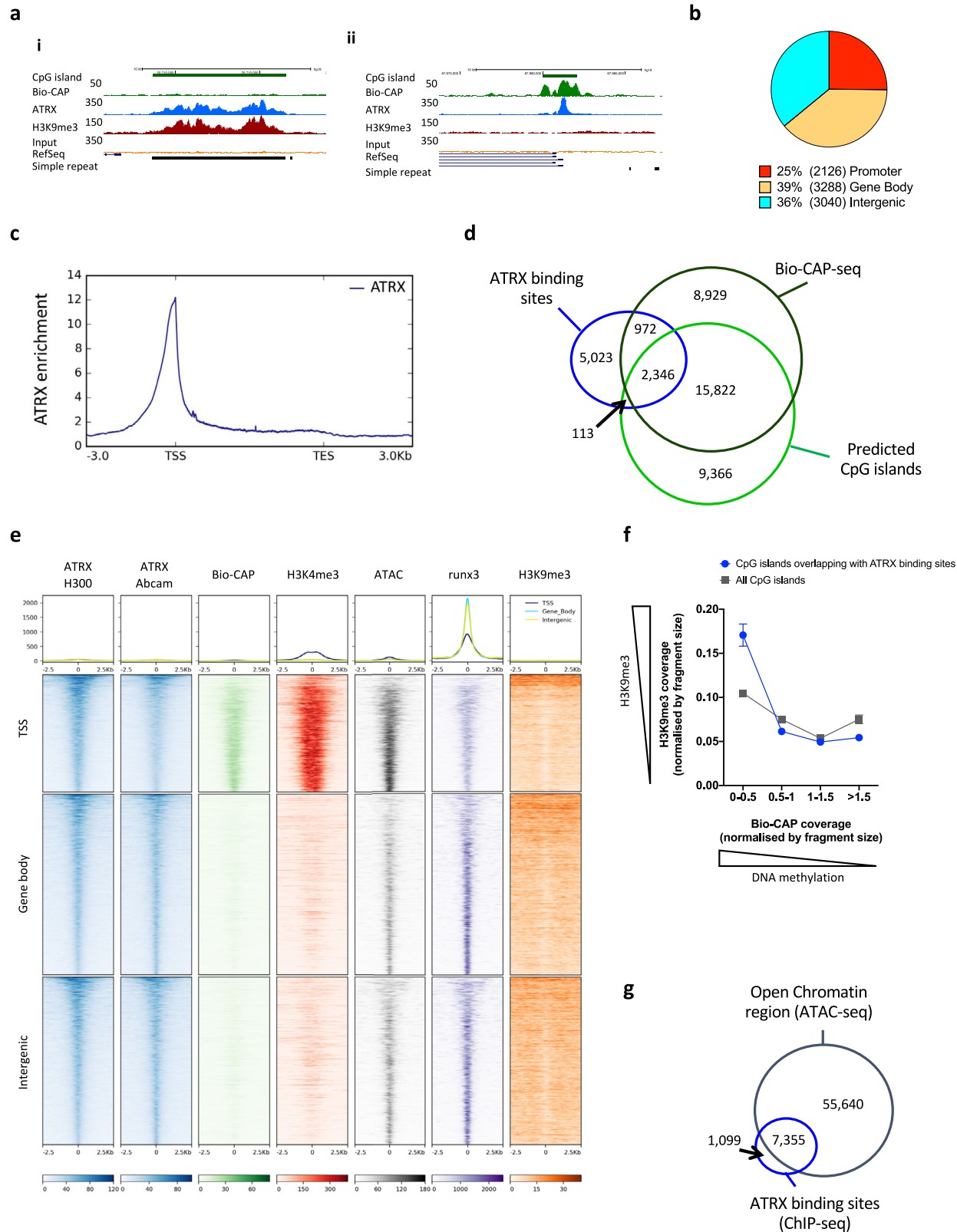

regions of open chromatin (*p*-value < 2.2e−16, odds ratio = 108.9, Fisher's exact test) (Supplementary Fig. 3d).

**ATRX binding sites are associated with active regulatory elements**. To determine the nature of the ATRX binding sites, the data were annotated with Genomic STate ANnotation

(GenoSTAN)[20]. Exploring 77,597 sites encompassing the open chromatin regions identified by ATAC-seq and ATRX binding sites, we identified seven chromatin states (active promoter (P), poised Promoter (Pp), active enhancers (E), enhancer-CTCF binding site (EC), CTCF site (C), repressed region (R) and background (B)) using this unsupervised approach (Fig. 2a and Supplementary Fig. 5). Of these, we observed a significant

**Fig. 1 ATRX binding sites associated with unmethylated CpG clusters and open chromatin regions. a** Examples of the variability in pattern and shape of the ATRX binding sites. (i) Example of a broad ATRX binding site enriched in H3K9me3 at a fully methylated CpG island with a background signal in the Bio-CAP experiment (designed to detect unmethylated CpG clusters)[17], (ii) Example of a sharp ATRX binding site depleted in H3K9me3 at a non-methylated CpG island with an enrichment in the Bio-CAP-seq signal. The signals represent an average of the independent replicates ($n = 3$) **b** Distribution of the ATRX binding sites. **c** Genome-wide analysis of ATRX enrichment across genes. The ATRX enrichment represents an average of the independent replicates ($n = 3$). **d** Overlap between ATRX peaks annotated CpG islands and non-methylated CpG clusters (Bio-CAP-seq) in LCLs. **e** Heatmap showing the enrichment in ATRX ($n = 3$), Bio-CAP ($n = 3$), H3K4me3 ($n = 3$), ATAC-seq ($n = 4$), Runx3 ($n = 2$) and H3K9me3 ($n = 3$) signals at all ATRX binding sites subdivided on their genomic location. The read densities represent an average of the independent replicates. **f** Correlation between Bio-CAP signal ($n = 3$ independent replicates) and H3K9me3 signal ($n = 3$ independent replicates) at all annotated CpG islands ($n = 27711$) in grey and at annotated CpG islands enriched in ATRX ($n = 2580$) in blue (data are presented as mean values $+/-$ SEM). **g** Venn diagram illustrating the overlap between all ATRX binding sites and ATAC-seq peaks.

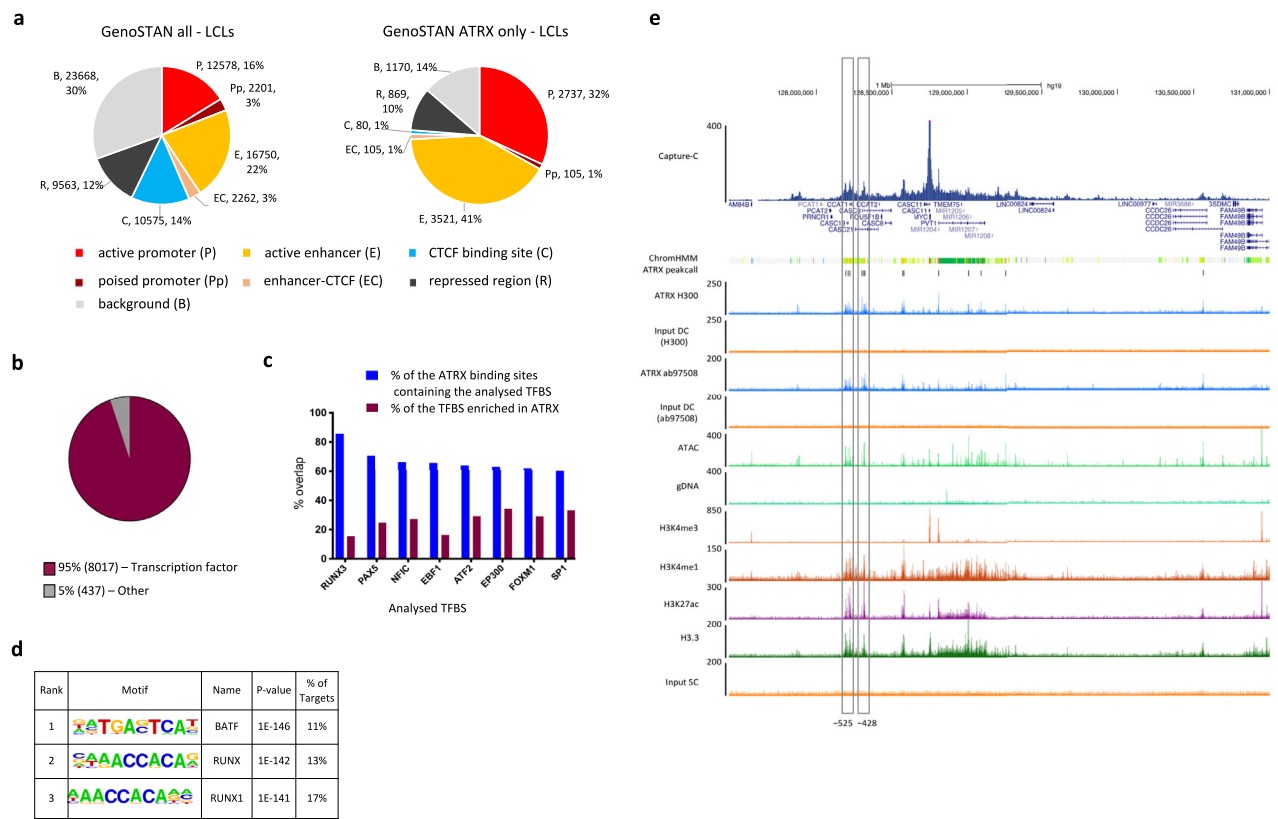

**Fig. 2 ATRX binding sites associate with active regulatory elements. a** GenoSTAN analysis in LCLs of: (left) all ATAC-seq sites of open chromatin identified in LCLs plus ATRX binding sites not in open chromatin, (right) all the ATRX binding sites identified in LCLs—active promoter (P), poised promoter (Pp), active enhancer (E), enhancer-CTCF binding site (EC), CTCF binding site (C), repressed region (R) and background (B). **b** Proportion of ATRX binding sites overlapping with transcription factor binding sites (TFBS). **c** Analysis of TFBS at ATRX binding sites (in blue) vs ATRX binding sites at TFBS (in burgundy) based on their % overlaps showing the TFBS the most enriched at ATRX binding sites. **d** Motif analysis of ATRX binding sites in LCLs (p-values HOMER findMotifsGenome.pl)[58]. **e** Representative image of the *Myc* locus highlighting the presence of ATRX enrichment at active enhancers (−525 and −428). The signals represent an average of the independent replicates ($n = 3$ for ATRX, H3K4me1, H3K4me3 and H3.3 ChIP-seq, $n = 2$ for H3K27ac ChIP-seq and $n = 4$ for ATAC-seq).

enrichment of ATRX binding sites at active enhancers (41% vs 22% of all sites, *p*-value < 2.2e−16, odds ratio = 2.5, Fisher's exact test) and promoters (32% vs 16% of all sites, *p*-value < 2.2e−16, odds ratio = 2.4, Fisher's exact test) (Fig. 2a and Supplementary Fig. 5c). These annotations were supported by analysing the chromatin state segmentation by Hidden Markov Model (ChromHMM) from an ENCODE LCLs dataset[21] (Supplementary Fig. 6a). Consistent with most ATRX binding at active regulatory elements, 95% of the ATRX binding sites contained at least one transcription factor binding site (TFBS) (Fig. 2b) (*p*-value < 2.2e−16, odds ratio = 141, Fisher's exact test compared to matching size random fragments) (Supplementary Fig. 3e)

with an over representation of RUNX3 in the LCLs analysed (present in 85% of ATRX binding sites (Fig. 2c)). Motif analyses supported an enrichment of RUNX family motifs at ATRX binding sites in regions of open chromatin in LCLs (Fig. 2d and Supplementary Fig. 6b), further highlighted with the metagenomic analysis (Fig. 1e).

Chromosome conformation capture (Capture-C) which identifies interactions between enhancers and promoters, was used to confirm the identity of gene-specific enhancers. We confirmed that ATRX signals at interacting promoters and enhancers were not due to artefacts caused by the binding of proteins at one site brought into close proximity and crosslinked to another site. In

particular, we observed loci in which only one of two interacting regulatory elements was enriched in ATRX (Supplementary Fig. 6d). As an example of enhancer-promoter interactions, we used the *MYC* promoter as viewpoint, and highlighted a complex pattern of enhancer-promoter interactions including those in which the regulatory elements are enriched in ATRX (Fig. 2e). These sites were also enriched in active chromatin marks including open chromatin regions identified by ATAC-seq, and histone modifications such as H3K27ac or H3K4me1. Notably, these ATRX binding sites also displayed an enrichment in H3.3 and were annotated as active enhancers and promoters by ChromHMM performed in LCLs[21] (orange in Fig. 2e). Together, these results confirm the interactions between ATRX and active promoters and enhancers and the associated chromatin modifications.

**ATRX binds to regulatory elements in a tissue-specific manner.** To determine the distribution of ATRX in a different cell type and compare this with the distribution in LCLs, ATRX ChIP-seq was performed in erythroblasts using differentiated CD34$^+$ human stem and progenitor cells (HSPCs) from normal human donors (Supplementary Fig. 7). Using our improved protocol and stringent peak calling, 12,659 ATRX binding sites were detected, a number more than ten times higher than previously described in erythroid cells[3]. As observed in LCLs, ATRX binding sites were mainly intragenic with a significant enrichment at TSS compared to matching size random fragments ($p$-value < 2.2e−16, odds ratio = 57.8, Fisher's exact test) and overlapping with open chromatin in erythroblasts ($p$-value < 2.2e−16, odds ratio = 138.5, Fisher's exact test) (Fig. 3a and b and Supplementary Fig. 3f and g). However, in erythroid cells, motif analysis showed that ATRX binding sites were enriched in GATA motifs in contrast to RUNX motifs in LCLs (Fig. 3c). ChIP-seq for histone marks and ATAC-seq showed that ATRX binds to active regulatory elements, confirming the similar observations in LCLs (Supplementary Fig. 8).

α-globin is a well-characterized, erythroid-specific target of ATRX[3]. Previous studies have identified ATRX predominantly at the G-rich ψζ variable number tandem repeat (ψζ VNTR) within the α-globin gene cluster. Here we found that ATRX is also significantly enriched at the CpG islands associated with the *HBA2* and *HBA1* genes and at the enhancers (R1, R2 and R4 in Fig. 3d). All of these sites showed the epigenetic features associated with active chromatin including chromatin accessibility, and enrichment in H3K4me3, H3K4me1 and/or H3K27ac (Fig. 3d). Furthermore, H3.3 enrichment was also observed at ATRX binding sites at *HBA1*, *HBA2* and their major enhancers R1 and R2 and ATRX enrichment in this region was only seen in erythroid cells when the elements are active and the elements and genes are transcribed. In contrast, H3K27me3, a repressive mark associated with α-globin silencing was depleted in erythroid cells but enriched across this locus in LCLs (Fig. 3d). These observations at the α-globin cluster confirm that ATRX interacts with active regulatory elements specifically in cells in which these genes are transcribed and suggest that its binding depends on the transcriptional activity of such sequences.

**ATRX binding is associated with changes in chromatin accessibility and gene expression.** We further investigated the characteristics of ATRX binding sites shared in both erythroid and non-erythroid (LCLs) cell types. We identified 2251 ATRX binding sites in LCLs (27%) overlapping with ATRX peaks in erythroblasts (Fig. 3e). As expected, shared peaks were enriched for gene promoters, CpG islands, open chromatin and PQS ($p$-value < 2.2e−16, odds ratio = 73.9, $p$-value < 2.2e−16, odds

ratio = 83.9, $p$-value < 2.2e−16, odds ratio = 50.7 and $p$-value < 2.2e−16, odds ratio = 7.2, respectively, Fisher's exact test, compared to matching size random fragments) (Fig. 3f–h and Supplementary Fig. 3h–k), supporting the hypothesis that ATRX binding/recruitment occurs in such regions of the euchromatic genome. Almost 95% of ATRX binding sites found in both LCLs and erythroblasts showed similar chromatin accessibility in both cell types (Fig. 3g). Interestingly, most of the open chromatin sites were found in both LCLs and erythroblasts contrasting with the matching size random fragments in which the majority of the open chromatin regions were cell type specific ($p$-value < 2.2e−16, odds ratio = 33.1, Fisher's exact test) (Supplementary Fig. 3j). As previous work has linked the recruitment of ATRX to transcription[3,22], the relationship between these factors was further examined by comparing ATRX enrichment and gene expression across both cell types. ATRX signals in LCLs were visualised according to gene expression in four classes. First, genes are expressed in both LCLs and erythroid cells; second, genes only expressed in LCLs; third, genes only expressed in erythroid cells, and fourth, genes are repressed in both cell types. As shown previously, ATRX enrichment at TSS is associated with transcription (Fig. 3i). Furthermore, we observed that ATRX enrichment at the TSS was correlated with the level of gene expression in LCLs (Fig. 3j). Our results thus show that enrichment of ATRX at gene regulatory regions correlates not only on their chromatin status but also on the level of gene expression.

**Pathogenic ATRX mutations are associated with changes in the chromatin environment at active regulatory elements.** To date, only a small number of genes have been identified that are affected by pathogenic ATRX mutations in humans. We performed microarray experiments on 20 LCLs from normal individuals and 28 LCLs from individuals with diverse pathogenic ATRX mutations, to identify more genes regulated by ATRX. When possible, we selected control-case pairs composed of ATR-X cases (all males) and their unaffected fathers thus reducing background effects (Supplementary Fig. 9). In total, 388 probes displayed an adjusted $p$-value < 0.05 (red dots in Fig. 4a), representing 234 differentially expressed genes (DEGs). The previous work[3] had shown that *NME4* (a nucleoside diphosphate kinase) was downregulated in LCLs derived from ATR-X cases and that this was correlated with the size of an intragenic G-rich tandem repeat which is a PQS. This downregulation was replicated in the current study (logFC = −1.34 compared to control) (Fig. 4a) and ATR-X cases displayed widely spread signals compared to controls (Supplementary Fig. 10a). Microarray data were further validated by qRT-PCR on 20/21 candidate genes including 17/18 DEGs (Supplementary Fig. 10b). Gene ontology analyses revealed a significant enrichment in biological processes linked to transcription (Table 1). Expression data were cross-referenced with the ATRX ChIP-seq data to analyse the distribution of the ATRX binding sites in the environment of the DEGs. Considering the complete set of DEGs and ATRX binding sites, we found a total of 83 DEGs containing at least one intragenic ATRX binding site and a total of 98 DEGs (10 kb window), 128 DEGs (50 kb window) and 156 DEGs (100 kb window) with at least one ATRX binding sites within these windows ($p$-value = 3.454e−06, odds ratio = 1.95, $p$-value = 1.512e−05, odds ratio = 1.81, $p$-value = 0.009595, odds ratio = 1.41, $p$-value = 0.01836, odds ratio = 1.39, respectively, Fisher's exact test) that could include genes directly affected by ATRX mutations (Supplementary Fig. 10c).

By way of example and supporting a role in the maintenance of chromatin integrity at regulatory elements, we observed that ATRX is enriched at the TSS of the following downregulated genes in ATR-X cases: the activating transcription factor *ATF7IP2*

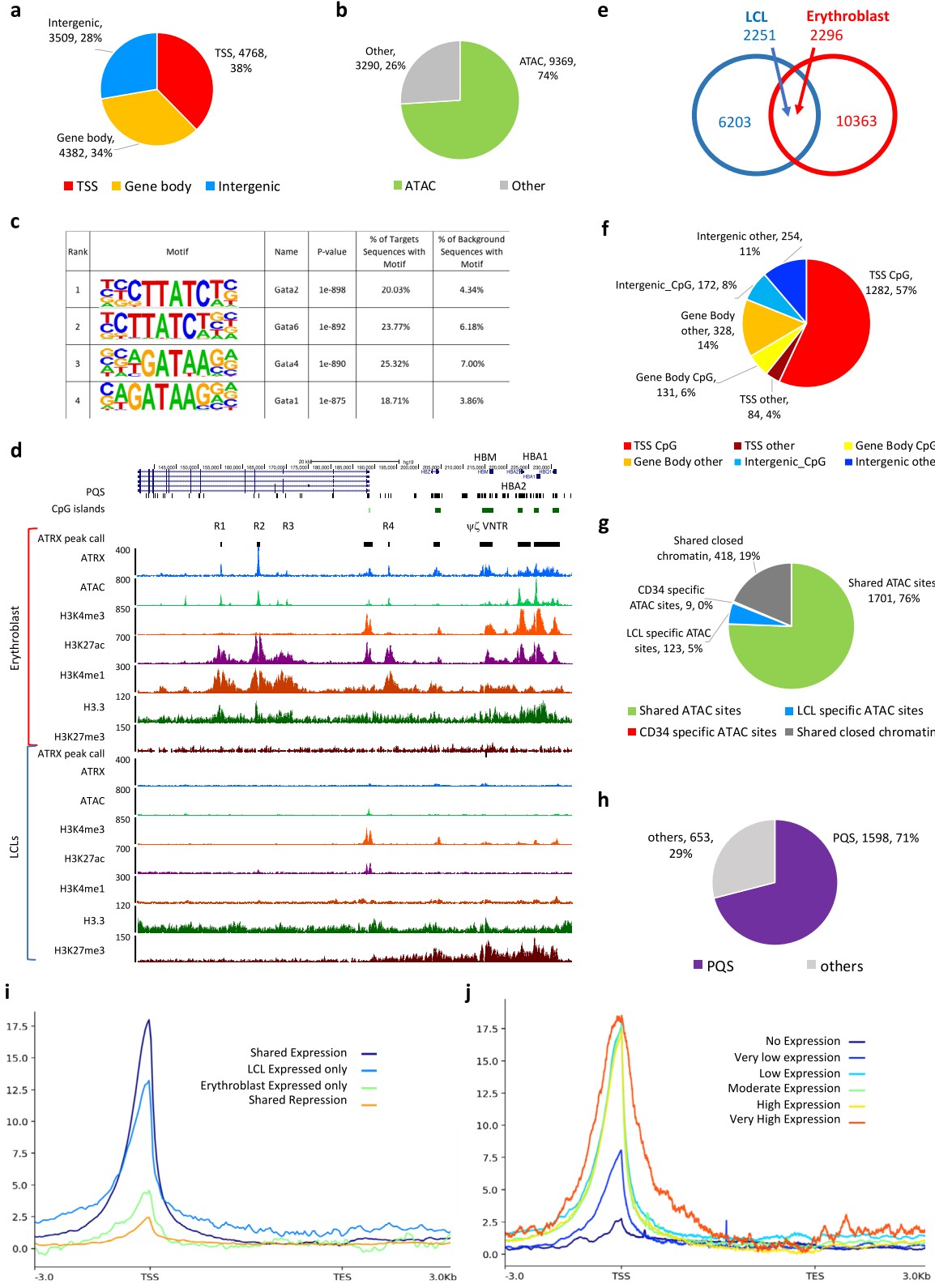

and three ZNFs (*ZNF555*, *ZNF57* and *ZNF718)*. Their promoter regions were also characterised by the presence of CpG islands and PQS (Fig. 4b, Supplementary Fig. 11a and b). At these loci, ATRX deficiency was associated with a change in the chromatin environment including a reduction in chromatin accessibility; a reduction in active chromatin marks including H3K4me3 and H3K27ac; reduced RUNX3 binding; a reduction in H3.3 at the

promoter and decreased H3K36me3, a mark associated with transcriptional elongation.

In addition, we observed hypermethylation at the *ATF7IP2* and *ZNF718* CpG island promoters in LCLs from ATR-X cases supporting previous data performed in peripheral blood cells from ATR-X cases[23] (Fig. 4b, Supplementary Figs. 11b and 12a). By contrast, the methylation status of the *ZNF555* and *ZNF57*

**Fig. 3 ATRX enrichment at regulatory elements varies with the activity of the locus and ATRX binding sites conserved across cell lines show conserved chromatin accessibility states. a** Distribution of the ATRX binding sites relative to genes in erythroblasts. **b** Distribution of the ATRX binding sites depending on their chromatin accessibility as assessed by ATAC-seq in erythroblasts. **c** Motif analysis of ATRX binding sites in erythroblasts (p-values HOMER findMotifsGenome.pl)[58]. **d** Representative image of the α-globin locus active in erythroblasts and silenced in LCLs. The signals represent an average of the independent replicates (in LCLS, $n = 3$ for ATRX ChIP-seq, H3K4me1, H3K4me3, H3K27me3 and H3.3 ChIP-seq, $n = 2$ for H3K27ac ChIP-seq and $n = 4$ for ATAC-seq. In erythroblasts, $n = 3$ for ATRX-ChIP-seq, $n = 1$ for H3K4me1, H3K4me3, H3.3 and H3K27me3 and $n = 4$ for ATAC-seq). **e** Venn diagram showing the cell type-specific and conserved ATRX binding sites in LCLs and erythroblasts. **f** Distribution of the conserved ATRX binding sites based on their position in relation to genes. **g** Chromatin accessibility status in LCLs and erythroblasts at the conserved ATRX binding sites. **h** Distribution of the conserved ATRX binding sites based on the presence or absence of PQS. **i** Genome-wide ATRX enrichment across genes in LCLs depending on the gene expression status in LCLs and erythroblasts. The signals represent an average of the independent replicates ($n = 3$). **j** Genome-wide ATRX enrichment across genes depending on the level of gene expression in LCLs. The signals represent an average of the independent replicates ($n = 3$).

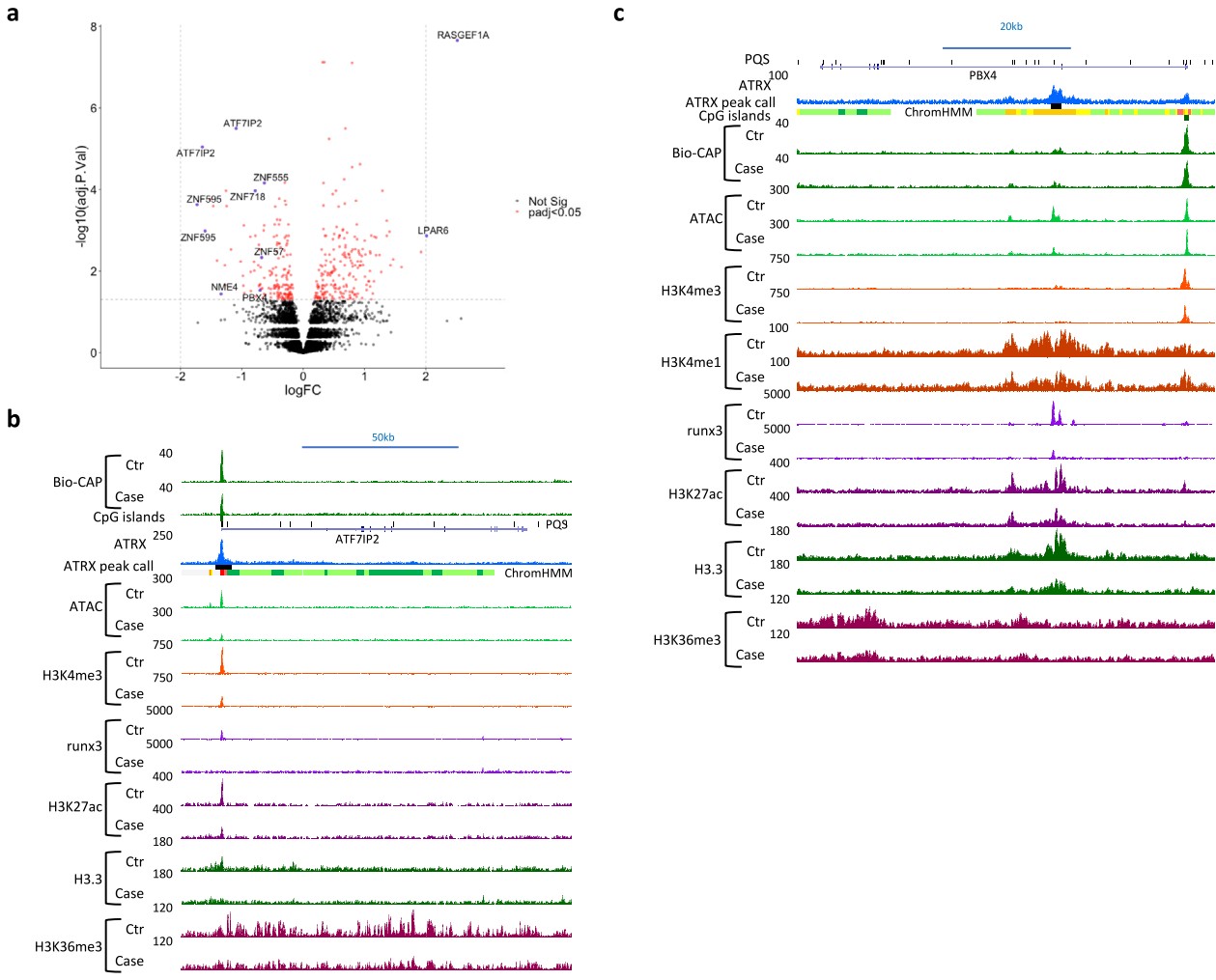

**Fig. 4 Pathogenic ATRX mutations are associated with changes in chromatin environment at regulatory elements in LCLs. a** Volcano plot of the microarray data comparing LCLs derived from ATR-X cases and unaffected donors. In red (or blue if highlighted with gene name), the probes that are significantly differentially expressed (ATR-X cases relative to controls) with an adjusted *p*-value ≤ 0.05 (horizontal grey dot line). adj.*P*.Val = adjusted *p*-values based on lmFit and eBayes (empirical Bayes moderated *t*-statistics test) from the limma packages[39] and adjustment with Benjamini Hochberg. Adjusted *p*-values for probes highlighted in blue: *ATF7IP2* (adj.*P*.Val = 9.14E−06 and 3.19E−06), *LPAR6* (adj.*P*.Val =1.37E−03), *NME4* (adj.*P*.Val = 3.67E −02), *PBX4* (adj.*P*.Val = 2.96E−02), *RASGEF1A* (adj.*P*.Val = 2.22E−08), *ZNF555* (adj.*P*.Val = 6.96E−05), *ZNF57* (adj.*P*.Val = 4.65E−03), *ZNF595* (adj.*P*.Val  = 2.36E−04 and 1.05E−03) and *ZNF718* (adj.*P*.Val = 1.08E−04). logFC = log fold change of cases over controls ($n = 20$ for the controls and $n = 28$ for the cases). **b** *ATF7IP2* locus. **c** *PBX4* locus. Signals of chromatin marks, Bio-CAP, ATAC, and Runx3, for ATR-X individuals (Case) and controls (Ctr). The signals represent an average of the independent replicates ($n = 3$ except for ATAC ($n = 4$) and H3K27ac and runx3 ($n = 2$) and ATRX ($n = 6$)).

CpG islands remained unaffected in ATR-X cases (Supplementary Figs. 11a and 12b) suggesting that DNA methylation is not the primary event affecting the chromatin environment in these cases.

Perturbation of chromatin was also observed at putative enhancers as illustrated at the *PBX4* locus (Fig. 4c). This gene, encoding a putative transcription factor, was downregulated in ATRX deficient cells and contained an ATRX binding site within

**Table 1 Gene ontology analysis based on the microarray data analysis from LCLs, ATR-X cases vs unaffected donors showing the top enriched pathways based on the count with a $p$-value < 0.05 ($p$-values based on EASE Score DAVID 6.8)[60].**

| Gene list | Term | p-value | Count | Genes |
|---|---|---|---|---|
| DEGs $p < 0.05$ | Transcription, DNA-templated | 0.042 | 32 | ZNF555, ZNF57, ZNF347, CNOT6, ZNF32, ZBTB38, ZNF30, HOXA1, LPXN, STAT4, LBH, TEAD4, VPS36, ZNF595, TXNIP, MBD3L3, L3MBTL4, TRNP1, ZNF506, ARID5A, ZNF649, MAPK11, MBD2, ZNF626, MXD1, ATF7IP2, PURA, MSL3, ZNF718, ADNP2, MAFIP, ZNF418 |
| | Regulation of transcription, DNA-templated | 0.025 | 27 | ZNF555, ZNF57, ZNF347, CNOT6, ZNF32, ZNF30, HOXA1, LPXN, STAT4, MEIS2, LBH, MYB, VPS36, ZNF595, L3MBTL4, TRNP1, ZNF506, ZNF649, ZNF626, ATF7IP2, MSL3, RPS6KA5, ZNF718, ADNP2, TCFL5, MAFIP, ZNF418 |
| | Angiogenesis | 0.045 | 7 | KLF5, FMNL3, NUS1, LEPR, SOX18, ENPEP, PNPLA6 |
| | T cell costimulation | 0.002 | 6 | BTLA, TRAC, CD80, KLRK1, KLRC4-KLRK1, CD28 |
| | Peptidyl-tyrosine dephosphorylation | 0.006 | 6 | PTPRC, PTPRG, DNAJC6, PTPDC1, PTPRO, RNGTT |
| | Negative regulation of gene expression | 0.021 | 6 | SLC35C2, BAK1, AIF1, TRIM6, CCR1, CD28 |
| DEGs $p < 0.05$ Intragenic ATRX binding site | Transcription, DNA-templated | 3.62E−04 | 20 | ZNF595, ZBTB38, MBD2, CTR9, ATF7IP2, ARID5A, ZNF32, ZNF57, CNOT6, LBH, ZNF718, ZNF649, ZNF418, ZNF506, ZNF626, ZNF30, ADNP2, ZNF347, ZNF555, L3MBTL4 |
| | Regulation of transcription, DNA-templated | 3.86E−05 | 19 | ZNF595, CTR9, ATF7IP2, ZNF32, ZNF57, MEIS2, CNOT6, LBH, ZNF718, ZNF649, ZNF418, ZNF506, ZNF626, ZNF30, MYB, ADNP2, ZNF347, ZNF555, L3MBTL4 |
| DEGs $p < 0.05$ ATRX binding site within 10 kb | Transcription, DNA-templated | 4.98E−04 | 22 | ZNF595, ZBTB38, MBD2, CTR9, ATF7IP2, ARID5A, ZNF32, ZNF57, CNOT6, LBH, ZNF718, ZNF649, ZNF418, ZNF506, ZNF626, ZNF30, STAT4, ADNP2, LPXN, ZNF347, ZNF555, L3MBTL4 |
| | Regulation of transcription, DNA-templated | 3.91E−05 | 21 | ZNF595, CTR9, ATF7IP2, ZNF32, ZNF57, MEIS2, CNOT6, LBH, ZNF718, ZNF649, ZNF418, ZNF506, ZNF626, ZNF30, MYB, STAT4, ADNP2, LPXN, ZNF347, ZNF555, L3MBTL4 |
| DEGs $p < 0.05$ ATRX binding site within 50 kb | Transcription, DNA-templated | 6.26E-4 | 26 | ZNF595, CTR9, ATF7IP2, MBD3L4, MBD3L3, LBH, ZNF506, ZNF649, ZNF626, STAT4, ADNP2, LPXN, ZNF347, ZBTB38, MBD2, ARID5A, ZNF32, MSL3, ZNF57, CNOT6, ZNF718, ZNF418, ZNF30, TXNIP, ZNF555, L3MBTL4 |
| | Regulation of transcription, DNA-templated | 2.28E−04 | 23 | ZNF595, CTR9, ATF7IP2, ZNF32, MSL3, ZNF57, MEIS2, CNOT6, LBH, ZNF718, ZNF649, ZNF418, ZNF506, ZNF626, ZNF30, MYB, STAT4, ADNP2, LPXN, ZNF347, ZNF555, L3MBTL4, TCFL5 |
| | Negative regulation of transcription from RNA polymerase II promoter | 0.044 | 10 | ZNF649, MBD2, MYB, CTR9, TXNIP, MBD3L4, MBD3L5, MBD3L2, MBD3L3, MEIS2 |
| DEGs $p < 0.05$ ATRX binding site within 100 kb | Transcription, DNA-templated | 0.0049 | 27 | ZNF595, CTR9, ATF7IP2, MBD3L4, MBD3L3, LBH, ZNF506, ZNF649, ZNF626, STAT4, ADNP2, LPXN, ZNF347, ZBTB38, MBD2, ARID5A, ZNF32, MSL3, ZNF57, MAPK11, CNOT6, ZNF718, ZNF418, ZNF30, TXNIP, ZNF555, L3MBTL4 |
| | Regulation of transcription, DNA-templated | 0.0032 | 23 | ZNF595, CTR9, ATF7IP2, ZNF32, MSL3, ZNF57, MEIS2, CNOT6, LBH, ZNF718, ZNF649, ZNF418, ZNF506, ZNF626, ZNF30, MYB, STAT4, ADNP2, LPXN, ZNF347, ZNF555, L3MBTL4, TCFL5 |

its gene body. In control cells, these ATRX binding sites showed marks of active enhancers (open chromatin, RUNX3, H3K4me1 and H3K27ac) and enrichment in H3.3 (Fig. 4c). Capture C confirmed a specific interaction between this site and the *PBX4* promoter, which is consistent with a role as an enhancer of *PBX4* (Supplementary Fig. 13). Strikingly, ATRX deficient cells displayed a decrease in chromatin accessibility, active chromatin marks (H3K27ac and H3K4me1) and H3.3 signals and a marked diminution in RUNX3 binding at this putative enhancer. Together, these results show that ATRX plays a role in initiating and/or maintaining chromatin accessibility and transcription factor occupancy at a subset of regulatory elements.

**Altered chromatin accessibility associated with a reduced α-globin expression in ATR-X syndrome.** By contrast with the common types of α-thalassaemia, in individuals with ATR-X syndrome and α-thalassaemia, the α-globin cluster is structurally intact. In these individuals downregulation of genes in the α-globin cluster (*HBA1/2* and *HBM*) results from a deficiency in ATRX. The previous work[3] showed that the length of the adjacent G-rich ψζ VNTR accounts for almost 60% of the variance in the severity of the α-thalassaemia as reflected in the number of red cells exhibiting HbH ($\beta_4$) inclusions. The mechanism by which the interaction between ATRX/DAXX/H3.3 and these upstream PQS repeats affect α-globin gene expression is unknown. However, this alone is insufficient to fully explain the phenotype, suggesting that a deficiency in ATRX downregulates α-globin expression via more than one of its various nuclear activities.

To investigate this further, we isolated CD34$^+$ HSPCs from two brothers diagnosed with the ATR-X syndrome (case 1 and case 2). These individuals had the same ATRX mutation but only one of them (case 1) had a detectable α-thalassaemia phenotype with 3% of red cells exhibiting HbH ($\beta_4$) inclusions.

CD34$^+$ HSPCs cells from these ATR-X individuals differentiated along the erythroid pathway in a similar manner to normal controls[24]. Due to the limited amount of material from these ATR-X cases, we prioritized the analysis of chromatin accessibility and assessment of H3K27ac, the most perturbed epigenetic features in ATR-X LCLs. In the affected individuals, we observed a depletion of H3K27ac and chromatin accessibility at *HBM*, the *HBA1/2* loci and the R1 enhancer (Fig. 5a). In addition, we showed that binding of the erythroid-specific transcription factor GATA1 was reduced at the ATRX bound R1 enhancer in the affected individuals. Together, these findings confirmed that ATRX deficiency was associated with changes in chromatin accessibility at active regulatory elements in this model locus (Fig. 5a). Capture C with the *HBA* promoters as a viewpoint did not show any drastic differences between affected individuals and normal controls suggesting that ATRX deficiency does not affect gene expression via significant changes in the 3D organisation of the locus (Supplementary Fig. 14). Consequently, it appears that changes in chromatin accessibility and associated chromatin modifications of regulatory elements within the α-globin locus are likely to contribute to the α-thalassaemia phenotype.

**Analysing the mechanism by which ATRX regulates chromatin and gene expression in single cells.** Although deficiency of ATRX causes α-thalassaemia, the red cell phenotype is quite different to that seen in the common forms of this anaemia. In particular, in ATR-X syndrome, most red cells have a normal haemoglobin content and volume (mean corpuscular haemoglobin and mean corpuscular volume) whereas both parameters are uniformly reduced in all cells in the common forms of α-thalassaemia. Furthermore, we have found that individuals (such as case1 and case2) with exactly the same mutation in the

*ATRX* gene have variable degrees of α-thalassaemia. This suggests that α-globin gene expression may not be affected in the majority of erythroid cells in ATR-X syndrome and that the presence of α-thalassaemia may be variable even with the same underlying mutation.

To determine if there might be cellular heterogeneity associated with changes in chromatin accessibility we performed single-cell ATAC-seq (scATAC-seq) in erythroblasts. As in bulk ATAC-seq, in a composite of the single-cell signals we observed a reduction in chromatin accessibility across the *HBA1/2* loci and at the R1 enhancer in the ATR-X cases case1 and case2 (Fig. 5a and Supplementary Fig. 15a).

At the single-cell level, the chromatin accessibility in erythroid cells from ATR-X cases was not reduced at the *HBB* locus compared to controls ($p$-value = 0.9995, Wilcoxon tests), whereas it was at the *HBA* and *HBM* loci ($p$-value < 2.2e−16, Wilcoxon tests) (Fig. 5b–d). $\beta_4$ inclusions result from an excess of HBB relative to HBA expression. We therefore assess the chromatin accessibility at the *HBA* or *HBM* loci relative to *HBB* in each cell in affected individuals and normal controls via a t-SNE analysis. The results revealed that most of the cells clustered in a major population (consistent with our differentiation protocol (Supplementary Fig. 7)) and identified a subpopulation of cells which displayed low accessibility at the *HBA* and *HBM* loci despite their high chromatin accessibility at the *HBB* locus (encircled with a red line in Fig. 5e–h). This subpopulation was predominantly composed of cells from ATR-X cases ($p$-value < 2.2e−16, odds ratio = 1.6, Fisher's exact test) most of which were derived from the ATR-X case 1 with α-thalassaemia ($p$-value < 2.2e−16, odds ratio = 2.3, Fisher's exact test)). This contrasts with subpopulations where there is an association between the high levels of chromatin accessibility at all three loci (*HBA*, *HBM* and *HBB*) in both ATR-X cases and controls (pointed to by arrows in Fig. 5e±g, Supplementary Fig. 15b). The subpopulations which displayed high chromatin accessibility at the *HBB* locus also displayed high chromatin accessibility at the *SLC4A1* locus (also known as *Band3* or *CD233*), a marker which is upregulated in late erythroblasts (Supplementary Fig. 15c).

Subsequently, we performed single-cell RNA-seq (scRNA-seq) on cells from the ATR-X case with α-thalassaemia (case1) and one control sample (Ctr3), and analysed the expression of HBA, HBM and HBB in single cells from both samples. Similarly to the scATAC-seq analysis, we observed a large population containing most of the cells (Supplementary Fig. 16a–d). Cells were negative for CD34 (a marker of human stem and progenitors) and positive for GPA (a marker expressed during terminal erythroid differentiation, such as at day 10 of differentiation) (Supplementary Fig. 16a). By highlighting only the cells which have a signal above quantile 5, we observed that the cells at the left side of the main population (black arrow in Supplementary Fig. 16b) tended to be slightly less differentiated. By contrast, the cells on the right part of the main population (red arrow in Supplementary Fig. 16b) tend to be more differentiated based on the expression of *GPA*, *SLC4A1* and *HBB*. In agreement with a previous report focusing on a homogenous population of cells[25], we observed that the cells clustered mainly based on their cell cycle phase rather than cell identity (Supplementary Fig. 16c). These results identified the minor population as slightly more differentiated G1 phase cells (Supplementary Fig. 16a–c). Splitting the cells between control and case, highlighted the presence of cells from the ATR-X case with a high expression signal for *HBB* but a reduce signal for *HBA* and *HBM* (pointed out by a red arrow in Supplementary Fig. 16d).

Using a scatter plot approach allowed to deeper assess the correlation between the expression of *HBB* and *HBA/HBM*. In the control, *HBA* and *HBB* expression levels were proportional,

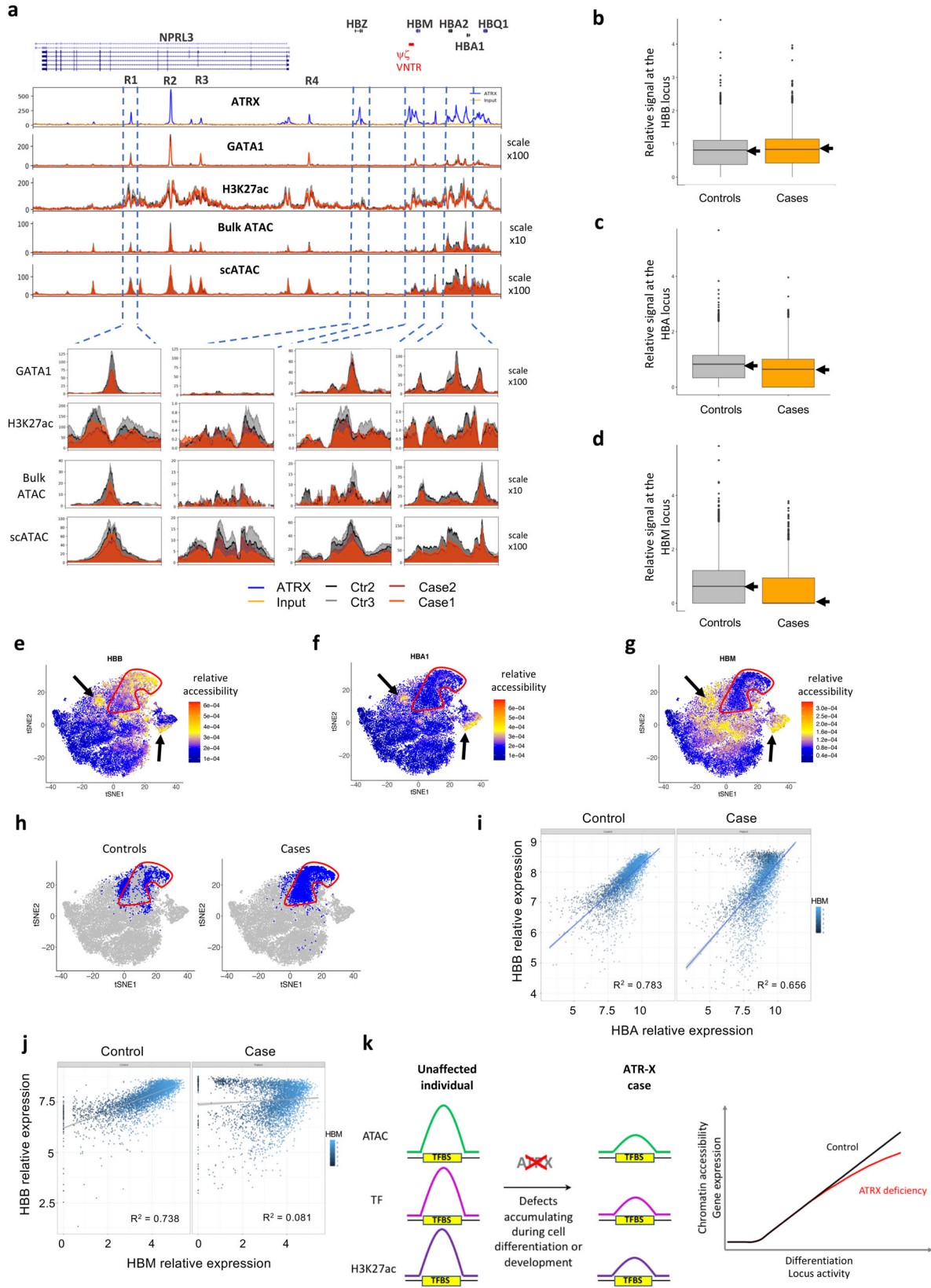

maintaining a constant *HBA*/*HBB* ratio between different cells throughout differentiation (Fig. 5i). Equivalent levels of *HBA* and *HBB* were also observed in partially differentiated ATR-X cells from case1 with low to medium *HBB* expression (i.e. relative expression of *HBB* below 8 in Fig. 5i). Conversely, in the most differentiated cells with a high level of *HBB* (relative expression

above 8), we detected a subpopulation of ATR-X cells with reduced *HBA* expression in case 1 (Fig. 5i). The reduced expression of *HBA* was supported by RNA-FISH data performed on erythroblasts from case 1 at day 10 and day 13 with the lower expression of *HBA* being most apparent at the later stages of differentiation (Supplementary Fig. 16e). Interestingly, ATR-X

**Fig. 5 ATRX loss of function is associated with perturbation of the chromatin environment and gene expression in a subpopulation of erythroblasts.**
**a** Comparison of the chromatin environment in erythroblasts across the α-globin locus between unaffected donors and ATR-X cases ($n = 1$). H3K27ac ChIP-seq were performed as ChIP-Rx and normalised based on the *Drosophila melanogaster* S2 cells spiked in. **b–d** Tukey based box plots of scATAC-seq data showing the distribution of chromatin accessibility in controls ($n = 2$: Ctr2 and Ctr3, each of them composed of 4000 cells) and ATR-X cases ($n = 2$: case1 and case2, each of them composed of 4000 cells). Tukey based box plots showing the 25th and 75th percentiles (lower and upper bounds of the box, respectively), the median (centre line highlighted by an arrow), the minimum value lower than the 25th percentile minus 1.5* inter-quartile range (IQR) (lower whisker) and the maximum value greater than the 75th percentile plus 1.5*IQR (upper whisker), any values beyond the whiskers boundaries are represented as dots. **b** *HBB* locus, **c** *HBA* locus and **d** *HBM* locus. **e–h** t-SNE analysis of scATAC-seq data showing each individual cell ($n = 4$: Ctr2, Ctr3, Case1 and Case2) and encircled in red a subpopulation with contrasting chromatin accessibility scores for **e** *HBB*, **f** *HBA1*, **g** *HBM*; in **h** highlighting the cells belonging to either controls or cases that are included in or surrounding the subpopulation encircled with a red line; in **e–g** the black arrows point to examples of subpopulations where there are high levels of chromatin accessibility at *HBB*, *HBA* and *HBM* loci. **i**, **j** scRNA-seq data showing each individual cell colour based on HBM expression in control ($n = 1$: Ctr3) and in case ($n = 1$: Case1) and showing in **i** the relative gene expression of HBA relative to HBB. and in **j** the relative gene expression of HBM relative to HBB. **k** Model of the loss of chromatin integrity observed in ATR-X cases during cell differentiation/ locus activity and the associated effect on gene expression.

---

cells with a relatively high level of *HBB* expression also showed an even more pronounced reduction in expression of the *HBM* gene (Fig. 5k) which lies close to the G-rich ψζ VNTR (Fig. 5a).

Taken together, our results suggest that when ATRX is mutated, changes in chromatin accessibility and gene expression predominantly affect a subset of late differentiating erythroid cells rather than causing a subtle change in the whole population (Fig. 5k). This provides an in vitro model recapitulating the α-thalassaemia associated with ATRX deficiency, and using this, we show that ATRX deficiency at α-globin and elsewhere in the genome, disrupts the epigenetic environment of active regulatory elements correlating with an impairment in gene expression.

## Discussion

It has previously been shown that the ATRX/DAXX/H3.3 complex binds repetitive sequences and putative G4 quadruplex forming sequences (PQS) in heterochromatin, where it plays an important ancillary role in multiple nuclear processes including replication, DNA repair, recombination, and transcription. Deposition of H3.3 in mammalian euchromatin has been generally considered to depend on the HIRA complex[26] although studies in drosophila and mouse have supported a role for ATRX in euchromatin[27,28].

Here using a substantially improved ChIP-seq protocol we have comprehensively and definitively shown that ATRX plays a role at multiple sites in euchromatic regions, in particular at enhancers and promoters including those located within the context of CpG islands. Importantly, many of these sequences are G-rich regions with the potential to form G4 quadruplex sequences, so called PQSs. The presence of ATRX at its binding sites in euchromatin is correlated with levels of histone H3.3, indicating that in addition to the HIRA complex, the ATRX/DAXX/H3.3 complex also plays a role in the deposition of H3.3 within euchromatic regions of the genome, in particular at active regulatory elements.

Analysis of the patterns of ATRX enrichment at regulatory elements in different cell types shows that recruitment of ATRX occurs in a tissue-specific rather than a gene-specific or transcription factor-specific manner. Furthermore, we show here that the level of ATRX enrichment is correlated with the level of gene expression. These findings are consistent with our previous observations that ATRX is preferentially recruited to actively transcribed regions[22]. In cells with loss of function mutations in ATRX, gene expression dysregulation is associated with variations in chromatin accessibility, active chromatin marks and H3.3 deposition at actively transcribed genes and their regulatory elements. At some loci, these changes predominantly affect promoters whereas at other loci enhancers were affected and, at both

promoters and enhancers, transcription factor binding is also reduced. Therefore it is possible that ATRX is required to maintain chromatin in an accessible state to facilitate a variety of nuclear processes during interphase by maintaining the presence or the positioning of the labile histone variant H3.3 at active genes and regulatory elements[29–31].

Methylation at CpG islands has previously been reported in blood cells from individuals with ATR-X syndrome[23]. We observe this at a subset of downregulated genes and it seems most likely that this CpG island methylation is secondary to other epigenetic changes rather than methylation changes dictating the epigenetic landscape at these loci. It has previously been shown that TF binding is necessary to prevent methylation of a CpG island promoter[32] and it is possible that at loci which become methylated when ATRX is mutated, chromatin remodelling by ATRX is necessary to facilitate TF binding[33] and keep these regions free of methylation.

α-thalassaemia is a hallmark of ATR-X syndrome and is due to the reduced expression of α-globin in erythroid cells. This is associated with a subsequent imbalance in globin chain synthesis and the production of excess β-globin chains leading to the formation of β-tetramers (HbH: $β_4$). This well-characterised locus provides an ideal model for understanding how ATRX normally plays a role in regulating gene expression. Here we have shown that in cells with a deleterious mutation in ATRX, chromatin accessibility at the α-globin cluster is reduced in only a subset of cells at intermediate and late stages of differentiation when the globin loci are most active. In single cells from this subgroup in which chromatin accessibility of α-globin is low, the accessibility of the β-globin gene is relatively high. This same subset of cells has an even greater reduction in the expression of the α-like gene *HBM*. These observations do not exclude the possibility that underlying defects associated with ATRX deficiency are present at earlier stages of differentiation but are below the level of detection and accumulate with rounds of cell division and differentiation.

Given the role of the ATRX/DAXX/H3.3 complex in facilitating a diverse range of nuclear processes, it is likely that any associated changes in gene expression due to the downregulation of components of this complex will result from different and even multiple effects. To address this we have used the human α-globin gene cluster as an experimental model.

Previously we have shown that the severity of α-thalassaemia in ATR-X syndrome is determined, to a large extent, (accounting for ~60% of the variance) by the length of a G-rich variable number tandem repeat (ψζ) lying upstream of the α-globin cluster to which ATRX binds. This G-rich sequence can form G4 structures in vitro and it has been speculated that the longer the repeat the more likely G4 is to form. ATRX binds to G4 in vitro[3] and in the absence of ATRX, G4 structures are more likely to form[34]

particularly when transcribed[22]. In the absence of ATRX, G4 structures might affect DNA replication through this PQS region and consequently perturb epigenetic memory, chromatin accessibility and gene expression[35–38]. The possibility of a replication fork meeting a G4 structure and the consequences of this would introduce a stochastic element to the effects on chromatin accessibility and gene expression. Single-cell analysis of erythroid cells from an individual with ATR-X syndrome strongly supports this notion. The proximity of *HBM* to the G4 forming ψζ repeat may explain the more severe downregulation that is observed at this locus.

The α-thalassaemia phenotype may not be fully explained by the effect of G-rich tandem repeat alone and it seems likely that the direct action of ATRX at the globin enhancers and promoters may also play a part in controlling α-globin expression by reducing chromatin accessibility and binding of transcription factors. No change in enhancer-promoter interaction was detected in ATR-X erythroblasts but rather there was a clear reduction in chromatin accessibility, H3K27acetylation and GATA1 binding at R1, one of the major enhancers in the globin cluster. In future studies to determine the relative contribution of ATRX at the α-globin regulatory elements could be assessed by excising the G-rich tandem repeat in erythroblasts.

The general observations that when ATRX has downregulated the chromatin at sites normally bound by ATRX becomes less accessible, less enriched in H3.3, less bound by their cognate transcription factors and less transcribed, together with the finding that ~50% of these sites are putative G4 quadruplexes, suggests that the mechanisms underlying downregulation of the α-genes may also be relevant to many other targets in euchromatin. These mechanisms together or independently may therefore explain many of the general effects of ATRX deficiency on gene expression at other loci.

## Methods

**Ethical compliance**. Use of blood cells from human participants was approved under REC reference: 07/MRE00/70 by Scotland A Research Ethics Committee. Informed consent was obtained for all participants. The study was authorized by the UK Medical Research Council, and the study design and conduct complied with all relevant regulations regarding the use of human study participants and was conducted in accordance to the criteria set by the Declaration of Helsinki.

**Cell culture**. LCLs derived from ATR-X cases and healthy controls were stored in liquid nitrogen in 10% Dimethyl-Sulphoxide (DMSO). Upon recovery, LCLs were thawed and washed with RPMI medium supplemented with 15% FCS, 1% Pen-Strep and 1% glutamine. Then, LCLs were incubated with supplemented RPMI media at 37 °C and 5% $CO_2$ until reaching the desired number which depended on the experiment.

CD34+ HSPCs were isolated from either 50 mL of healthy adult peripheral blood or leucocyte reduction filters (obtained from the NHSBT) using Histopaque-1077 Hybri-Max (Sigma) density centrifugation and positive bead selection (130-046-702, CD34 Microbead Kit, Miltenyi Biotech Cat.). The CD34+ HSPC fraction was cryopreserved in freezing media consisting of 90% FBS (Gibco) and 10% DMSO.

Frozen CD34+ HSPCs were recovered into base medium (IMDM Source BioScience UK Limited) containing 3% (v/v) AB Serum, 10 μg/mL insulin, 3 U/mL heparin (all from Sigma-Aldrich, Poole, UK), 2% (v/v) foetal bovine serum (Gibco), 200 μg/mL Human holo-transferrin (R&D Systems Minneapolis, MN, USA). For the expansion phase of the three-stage erythroid culture system the media was supplemented with 3 U/mL EPO (Eprex Janssen Cilag UK), 10 ng/mL stem cell factor (SCF) and 1 ng/mL IL-3 (Peprotech, London UK). Cells were maintained at a density of $2 \times 10^5$ cells until day 7 when they were transferred to phase 2 differentiation media consisting of base media with 3 U/mL EPO and 10 ng/mL SCF. From this point, cells were staged using cytospins and FACS using a panel of 6 fluorophore-conjugated monoclonal antibodies (Supplementary Fig. 17). Staining was conducted in FACS buffer containing 10% FBS and 90% PBS (both from Gibco, UK) and samples analysed using the Attune NxT (Life Technologies, California USA). Gates were set using fluorescence-minus-one controls run for each differentiation and data analysed using FlowJo software v 10.0.8r1. Cells were collected at day 10.

**RNA extraction**. RNA was extracted using RNeasy Mini kit from Qiagen. $4 \times 10^6$ cells were used as starting material. Lysates were homogenised using QIAshredder spin column (79656, Qiagen). Samples were submitted to on-column DNase digestion using RNAse-Free set DNase Set (79254, Qiagen) and RNA quality assessed by Nanodrop and integrity was assessed using Agilent RNA ScreenTape assay system (5067-5576/7/8, Agilent). Only samples displaying a RINe score higher than 8 were selected. Note that RNA extractions for the microarray experiments were carried out in batches of 7–16 cell lines resulting in a potential batch effect for further analysis. All the samples present in the same batch were processed at the same time from the cell culture to the microarray experiment.

**Reverse transcription**. cDNA was produced using High Capacity RNA-to-cDNA Kit (4387406, Applied Biosystems) using 1.5 μg of RNA as starting material and following the manufacturer's instructions.

**Primeview affymetrix microarray**. The microarray experiment was performed using PrimeView™ Human Gene Expression Array Plate and following the 100 or 81/4-format 3′ IVT PLUS Reagent Kit User Manual (Affymetrix). 250 ng of RNA were used as the starting material. In short, RNA samples were used for cDNA synthesis followed by the production of biotinylated-labelled cRNA. Labelled cRNA was then purified, fragmented and hybridised on arrays. Quality controls were performed during the process using Agilent Technologies Agilent RNA ScreenTape System. After staining, the arrays were scanned. Results were further analysed using Affymetrix® Transcriptome Analysis Console (TAC) and Affymetrix® Expression Console™ Softwares (Affymetrix).

**Power calculation and correction for the batch effects in microarray experiment analyses**. The power calculation was determined using the samr and ssize packages from R. In addition to the microarray analysis as per the manufacturer's instruction (described above), a parallel analysis was performed allowing the correction of the batch effect generated during RNA extraction using R script with limma[39], affy[40] and genefilter[41] packages. To increase the power, half of the probes were removed after normalisation based on their very weak signal or weak variation across samples[42].

**High-throughput qRT-PCR by Fluidigm Biomark HD system**. High-throughput qRT-PCR was performed with preliminary extracted RNA using a Fluidigm 192:24 Gene Expression Chip and Taqman assay (ThermoFisher Scientific) listed in Supplementary Table 2. Data were analysed using the Fluidigm Real-time PCR analysis 3.1.3 software and were normalised to the mean of two housekeeping genes (*RPL13A* and *GAPDH*).

**Quantitative-PCR**. qPCR based on the Taqman assay was performed using Taqman probes with a 2x Taq mastermix (Applied Biosystems) and Taqman expression Assay (ThermoFisher Scientific) listed in Supplementary Table 2. qPCR experiments based on SYBR green assay were performed using SYBR green master mix (4309155, Applied Biosystems). The primers used for these experiments are listed in Supplementary Table 3.

**RNA-FISH**. Working in an RNase-free manner, cells were settled and fixed on poly-L-lysine coated slides in 4% formaldehyde/ 5% acetic acid for 20 min at room temperature. Following three washes for 5 min in 1× PBS, slides were stored in 70% ethanol at –20 °C until required. Prior to hybridization, slides were pre-treated with 0.02% pepsin for 5 min at 37 °C in 0.01 M HCl. Following a rinse in water, cells were fixed for 10 min in 3.7% formaldehyde at room temperature. Slides were then washed in 1× PBS, dehydrated consecutively in 70%, 90% and 100% ethanol and air dried. Hybridization was commenced overnight at 37 °C in hybridization mixture containing 1 ng/μl of each oligonucleotide probe in 25% formamide, 2× SSC, 200 μg/ml salmon sperm DNA, 5× Denhardt's, 1 mM EDTA and 50 mM sodium phosphate pH 7.0. Following hybridization, cells were washed three times for 10 min in 2× SSC at 37 °C, followed by a 5 min wash in 0.1 M Tris, 0.15 M NaCl, 0.05% Tween 20 at room temperature. Blocking of slides and antibody detection of the labelled probes were carried out in 0.1 M Tris, 0.15 M NaCl solution containing 1.35% (wt /vol) ELISA blocking reagent (11112589001, Roche). Digoxigenin (DIG)-labelled α-globin probes were detected with antibody layers of 1:50 sheep anti-DIG FITC (11207741910, Roche) and 1:100 rabbit anti–sheep FITC (FI-6000, Vector Laboratories) Slides were mounted in Vectashield mountant (Vector) with 1ug/ mL DAPI added. RNA FISH Probes: Human α-globin intronic probes: (GGAGCAGGGGAGGGAGCCTCACCTCTCCAG; GGCCAGGACGGTT-GAGGGTGGCCTGTGGGT; AAGAAGGCGCCATCTCGCCCCTCGACCCAG; AGGCCCGCAGCCGCCGCCTGCGCTACACCT).

**ATRX ChIP**. All ATRX ChIPs have been carried out following the optimised protocol described in ref. [16]. For ATRX ChIP-seq in LCLs, the experiment has been performed on three independent cell lines (FF-Ctr, TA-Ctr and CB-Ctr) using two different antibodies (Santa Cruz H300 sc-15408 (10 μg) and Abcam ab97508 (15 μg)). In brief, $4 \times 10^7$ to $10^8$ cells were crosslinked with 2 mM EGS at room temperature for 45 min and 1% formaldehyde for 20 min. Cells were lysed with

lysis buffer 1 (100 mM HEPES, 140 mM NaCl, 1 mM EDTA, 10% glycerol, 0.5% NP-40 and 0.25% Triton X-100), then with lysis buffer 2 (200 mM NaCl, 1 mM EDTA, 0.5 mM EGTA and 10 mM Tris pH 8) and finally with lysis buffer 3 (1 mM EDTA, 0.5 mM EGTA, 1 mM Tris–HCl pH 8, 100 mM NaCl, 0.1% sodium deoxycholate and 0.5% N-lauroyl sarcosine). Chromatin was sonicated, samples were pre-cleared with Protein A Dynabeads and incubated over night at 4 °C with Dynabeads pre-coated with ATRX antibody. Samples were washed five times with wash buffer (50 mM HEPES, 1 mM EDTA, 0.7% sodium deoxycholate, 500 mM LiCl and protease inhibitor cocktail) and once with ice-cold PBS before elution with 0.5% SDS and 0.1 M sodium bicarbonate, reverse crosslinking with 0.2 M sodium chloride at 65 °C for 4 h and DNA purification. For ATRX ChIP-seq in erythroblasts, the experiment has been performed on four samples of erythroblasts extracted from three donors (Supplementary Table 4).

**Transcription factors ChIP.** $10^8$ cells were crosslinked with 2 mM DSG for 30 min and 1% FA for 30 min. Cells were lysed (1% SDS, 10 mM EDTA, 50 mM Tris, pH 8.1(+ protease inhibitors) and sonicated on the Covaris to give fragments of 100–300 bp. Samples were diluted (0.01% SDS, 1.1% TritonX-100, 1.2 mM EDTA, 16.7 mM Tris-HCl, pH 8.1,167 mM NaCl (+ protease inhibitors) and incubated over night with 1 μL of Runx3 (39301 Active motif/ ThermoFisher) and 2 μL of Runx3 (653604 BioLegend) or 2 μL Recombinant Anti-GATA1 antibody (EPR17362) - ChIP Grade (ab181544, Abcam). Magnetic protein A and G beads (26162, ThermoFisher Scientific) were used to isolate antibody-chromatin complexes. Samples were washed three times with with RIPA buffer (50 mM Hepes-KOH, pH 7.6, 500 mM LiCl, 1 mM EDTA, 1% NP-40, 0.7% Na-Deoxycholate) and then with TE + 50 mM NaCl before elution (50 mM Tris-HCl, pH 8.0, 10 mM EDTA, 1% SDS) following by RNase and proteinase K treatment and DNA purification. For information regarding the replicates see Supplementary Table 4.

**Single step cross-linking ChIP.** The single step cross-linking ChIP-seq was performed using the ChIP Assay kit (Millipore) using the antibodies: 4 μg of H3.3 (09–838, Millipore/Merck), 3.5 μg of H3K27ac (ab4729, Lot: GR276932-1, Abcam), 3.5 μg of H3K9me3 (ab8898, lot GR3244171-1, Abcam), 3.5 μg of H3K4me3 (ab8580, lot: GR273043-1, Abcam), 3.5 μg of H3K4me1 (ab8895, lot: GR283603-1, Abcam), 10 μg of H3K27me3 (ab6002, lot: GR275911-6-1, Abcam), 10 μg of H3K36me3 (ab9050, lot: GR233723-2, Abcam), 3.5 μg of H3K4me3 (07-473, lot: 2664283, Millipore/Merc) and 3.5 μg of H3K4me1 (ab195391, lot: GR304893-2, Abcam). For ChIP-Rx, crosslinked *Drosophila melanogaster* S2 were added at the lysis step of the ChIP as described in[43] at a ratio of 1:4. Briefly, $2 \times 10^6$ to $10^7$ cells were crosslinked with 1% formaldehyde at room temperature for 10 min. Cross-linking was quenched with 130 mM glycine, and cells were washed with cold PBS. Fixed material was stored at −80 °C. Samples were lysed by incubation on ice with lysis buffer for 15 min. In samples noted as spiked in, fixed *Drosophila melanogaster* S2 cells were added. Lysed cells were sonicated on the Covaris S220. Insoluble material was removed by centrifugation and soluble material was diluted with dilution buffer. Immunoprecipitation was performed by overnight incubation of diluted chromatin with the antibodies mentioned above. Protein A and/or G agarose beads were added. After 1 h incubation at 4 °C, washes and elution were performed according to the manufacturer's instructions. DNA was purified by phenol-chloroform extraction and ethanol precipitation with NaOAc, and 2 μl GlycoBlue. For information regarding the replicates see Supplementary Table 4.

**Biotinylated CxxC affinity purification (Bio-CAP).** Bio-CAP experiments were performed as described in ref. [17]. In brief, genomic DNA was extracted from $2.5 \times 10^7$ cells using 950 μL of extraction buffer (20 mM Tris HCl (pH 8.0), 10 mM EDTA, 100 mM NaCl and 0.5% SDS) followed by RNaseA/T1 (EN0551, Thermo Scientific) treatment and phenol/chloroform extraction. gDNA were sonicated to an average size of 200 bp and diluted to 17.5 mg/mL in CAP100 buffer (12.5% glycerol, 0.1% Triton-x-100, 20 mM HEPES pH 7.9 and 100 mM NaCl). 100 μL aliquot was retained as input. NeutrAvidin Agarose Resin (29201, Thermo Scientific) was conjugated with biotinylated hKDM2B ZF-CxxC protein (kindly provided by Rob Klose). 500 μL of diluted sonicated DNA were incubated with conjugated CxxC resin for 1 h at 4 °C. Serial washes and elutions using increasing salt concentration were performed. First, two washes with 1 mL of CAP100 buffer were carried out, followed by two elutions with 50 μL of CAP300 (12.5% glycerol, 0.1% Triton-x-100, 20 mM HEPES pH 7.9 and 300 mM NaCl) and two elutions with 50 μL of CAP500 (12.5% glycerol, 0.1% Triton-x-100, 20 mM HEPES pH 7.9 and 500 mM NaCl). Subsequently, the elution process was repeated using buffers with 700 mM and 1 M NaCl sequentially. DNA purification from eluted samples and input were performed using QIAquick PCR purification kit (28104, Qiagen) and assessed by qPCR. Samples derived from elutions using 700 mM and 1 M NaCl were used for sequencing. Bio-CAP-seq experiment has been performed in three independent LCLs in controls and three independent LCLs in patients as three case-control pairs each of them being derived from the unaffected father (control) of an ATR-X patient (case).

**Assay for transposase-accessible chromatin with high-throughput sequencing (ATAC-seq).** ATAC-seq was carried out as described in ref. [44] using 80,000 cells as starting material. Cell pellets were first resuspended in 4 °C ATAC lysis buffer

(10 mM Tris-HCl, 10 mM NaCl, 3 mM MgCl2 and 0.1% (v/v) IGPAL) and washed once with PBS. After a 10 min centrifugation at 4 °C and 500 ×g, pellets were resuspended in a transposition reaction mix (Illumina Nextera DNA Library Prep Kit, FC-121-1030) for 30 min at 37 °C. As a control, transposition reactions were also performed on gDNA. Transposed DNA was extracted using a MinElute PCR Purification kit (28004, Qiagen) and PCR amplified and indexed. Indexed libraries were purified using QIAquick PCR purification kit (28104, Qiagen).

**Capture-C.** Capture-C experiment was performed following the protocol described in[45]. Briefly, cells were crosslinked with 2% formaldehyde (10 min, room temperature); quenched with cold glycine; washed in phosphate-buffered saline; resuspended in cold lysis buffer (tris 10 mM, NaCl 10 mM, NP40 0.2%, complete proteinase inhibitor (Roche)) and snap frozen to −80. Cells were thawed on ice, and washed in milliq DpnII buffer. Cells were then resuspended with 0.25% SDS and restriction enzyme buffer and incubated at 37 °C for 1 h at 1400 rpm on a Comfort Thermomixer (Eppendorf) followed by a further incubation of 1 h following the addition of triton X100 (final concentration 1.67%). Overnight digestion was performed using Dpn II (500U /ml (NEB) at 37 °C/1400 rpm). The digested chromatin was ligated overnight (Fermentas HC Ligase final concentration 10 U/ml) at 16 °C at 1400 rpm on the Thermomixer. Nuclei were purifed by centrifugation. The samples were then decrosslinked overnight at 65 °C with Proteinase K (Roche) followed by a 30 min incubation at 37 °C with RNAse (Roche). Phenol/Chloroform extraction was then performed followed by an Ethanol precipitation and a wash with 70% Ethanol. Digestion efficiencies were assessed by gel electrophoresis (1% agarose) and RT-PCR (Taqman), which showed digestion efficiencies in excess of 70%. DNA content of the Dpn II 3 C libraries were quantitated using a Qubit fluorometer (Life technologies). 5–10 μg of each library was sheared using a Covaris S2 in milliq dH2O. Covaris settings used were: duty cycle 10%, Intensity 5, Cycles/burst 200, Time 6 cycles of 60 s, set mode frequency sweeping temperature 4–7 degrees. Following shearing DNA was purified using AMPureXP beads (Agencourt) and DNA quality assessed. DNA end repair and adapter ligation were performed using the NEB Next or NEB Ultra II DNA sample preparation reagent kits, depending on the amount of DNA available, using the standard protocol. Biotinylated capture oligonucleotides were designed to the ends of the viewpoint fragments (list of Capture-C oligos in Supplementary Table 5). Where possible 1–2 μg of each adapter-ligated library were hybridized with the biotinylated capture oligonucleotides, using the Nimblegen SeqCap reagents and an adapted protocol. The quality of the resultant captured library was assessed by Agilent tape station or bioanalyser (D1000).

**Library preparation.** Unless specified otherwise, library preparation was performed using NEBNext Ultra DNA Library Prep Kit for Illumina (E7370L, NEB) and the NEBNext Multiplex Oligos for Illumina (E7335, NEB) following the manufacturer's protocol.

**Single cell ATAC-seq.** sc-ATAC-seq was performed using the 10X Genomics Chromium™ i7 Multiplex Kit N Set A (1000084), Chromium™ Chip E Single Cell ATAC Kit (1000086) and Chromium™ Single Cell ATAC Library & Gel Bead Kit (1000111) using 7000 nuclei/μl as a stock concentration following the manufacturers' instructions and sequenced as described below in Library quantification and High throughput sequencing. sc-ATAC-seq analysis was performed using the cellranger-atac workflow by first demultiplexing the Illumina® sequencer's base call files to create fastq files using cellranger-atac mkfastq. fastq files were mapped to the human genome hg19 and analysed using Cellranger-atac count. Further analysis was performed using the Seurat package and its Signac extension[46]. Low count cells (passed_filters > 500) were filtered and 4000 cells per sample were randomly selected. Normalised signals were compared by boxplots. In parallel, the samples were analysed with cisTopic[47] on aggregated samples with the following parameters; $\alpha = 50$, $\beta = 0.1$, iterations = 500 and a number of topics between 2 and 100 (2, 10, from 20 to 60, 1 by 1; from 70 to 100, 10 by 10). The chromatin accessibility around the *HBM*, *HBB* and *HBA* genes were analysed based on the gene score activity.

**Single cell RNA-seq.** scRNA-seq was performed using the 10X Genomics Chromium™ Single Cell 3′ Reagent Kits v3 (1000075) with 10,000 cells as starting material following the manufacturers' instructions and sequenced as described below in Library quantification and High throughput sequencing. Sequenced data were analysed using cellranger/3.0.2. The scRNA-seq data were then analysed using the Seurat package[48]. Cells with more than 7.5% of mitochondrial genes and as well as cells containing less than 800 nFeature_RNA or more than 4000 nFeature_RNA were excluded. A random subset of 4400 cells per sample were selected and the data were normalized using the log normalize method with a scale factor of 10,000. Normalised data were extracted from the Seurat object, data for HBA1 and HBA2 were added as sumHBA and scatter plots were performed using ggplot2.

**Library quantification and High throughput sequencing.** The sample size distribution was assessed using D1000 ScreenTape assays (5067-5582/3, Agilent) on the Agilent 2200 TapeStation system. Samples were quantified using the Qubit dsDNA HS Assay (Q32851) and KAPA Library Quantification Kits for Illumina®

platforms (KR0405, KAPA Biosystems). 4 nM libraries with compatible indexes were pooled and sequenced on a NextSeq 500 sequencer (Illumina) using either 300 cycles (Capture-C only), 150 cycles or 75 cycles NextSeq 500/550 High Output v2 kit (Illumina).

**Analysis of raw sequencing data from ChIP-seq, ATAC-seq and Bio-CAP-seq experiments.** Fastq files were aligned to the human genome hg19 using an in-house pipeline described in[49] (https://github.com/Hughes-Genome-Group/NGseqBasic/releases) using Bowtie 2[50]. In addition, –atac flag was used for ATAC-seq analysis. Data quality was assessed using fastQC reports[51]. Trimming of the adapter sequences was performed using trim-galore[52]. PCR duplicates were removed using Samtools[53]. Signal artefact blacklisted regions[54] were excluded. For normalisation, with the exception of ChIP-Rx data which were normalised following[43] (the normalization factor was adjusted to take into account the ratio of mapped human (hg19) and Drosophila (dm3) reads in the ChIP and input samples), the total number of mapped reads or a total number of mapped reads under peak regions was determined after the removal of PCR duplicates and excluding the reads mapping to the mitochondrial chromosome. Reads were normalised per 100 million reads. Data were visualised on the UCSC genome browser (http://genome.ucsc.edu/)[55]. Versions of software packages used for the analysis include FASTQC 0.11.9, Bowtie 2.3.2, Samtools 0.1.19, Bedtools 2.25.0 to 2.29.2, Deeptools 2.2.2 to 2.4.2, ucsctools 373.

**Capture-C probes design and data analysis.** Capture-C probes were designed using CapSequm (http://apps.molbiol.ox.ac.uk/CaptureC/cgi-bin/CapSequm.cgi). Capture-C data were analysed with an in-house pipeline CCseqBasic mapping to hg19[56] (GitHub: https://doi.org/10.5281/zenodo.4196777).

**Peak calling.** Peak calling was performed using MACS2[57] or HOMER[58] using input (for ChIP-seq and Bio-CAP-seq) as a background control. Each sample was independently peak called before filtering based on the representation of each peak across samples. Peaks were retained for the final peak call only if: (1) they were present in at least three samples (Bio-CAP-seq) and independent of the antibody (ATRX ChIP-seq in LCLs) or (2) they were present in at least two samples for the other experiments. Peak calling quality was assessed by direct visualisation on MIG[59] and peaks associated with artefacts were removed. An additional stringency threshold was applied by excluding any peaks associated with regions to which the input signal or ATAC-seq performed on gDNA was above the set threshold.

**Gene ontology and motif analyses.** Gene ontology analysis for the differentially expressed genes has been performed using GOTERM_BP_DIRECT DAVID 6.8[60].
Gene and genome ontology analyses based on the ATRX binding sites were carried out using HOMER annotatePeaks.pl[58].
Motif analysis of the ATRX binding sites was performed using HOMER findMotifsGenome.pl[58].

**Graphical analysis.** Heat maps and profile plot analyses were performed using Deeptools[61]. Volcano plots and single cell analysis plots were plotted on R. Box plots and associated statistical analyses were performed with GraphPad Prism 9 or R.

**Repeatome analysis.** Repeat enrichment in ATRX ChIP-seq data (read one of the paired end only) was performed using a repeat analysis pipeline based on[62], and mapped using the hg18 assembly to compare a union of the fastq reads of ChIP vs Input.

**RefSeq and CpG island annotation.** RefSeq annotation for hg19 was used to characterise the promoter, gene body and intergenic regions identified at ATRX binding sites[63]. The CpG island annotation for hg19 was retrieved from the cpgIslandExt table in the UCSC database.

**GenoSTAN analysis.** GenoSTAN analysis was performed using R scripts based on[20]. Analyses were performed on 1000 bp fragments covering the ATAC-seq peaks identified in LCLs and erythroblasts as well as the ATRX binding sites not overlapping with ATAC-seq peaks.
For the annotation in LCLs, H3K4me3, H3K4me1, H3K27ac, H3K27me3 and H3K27me3 ChIP-seq datasets were used. In addition, CTCF ChIP-seq (GEO accession number GSM733752) was used in the analysis. 12 states were initially used and analogous states were subsequently pooled together resulting in seven distinct states: active promoter (P), poised Promoter (Pp), active enhancers (E), enhancer-CTCF binding site (EC), CTCF site (C), repressed region (R) and background (B).
For the annotation in erythroblast, five chromatin marks, H3K4me3, H3K4me1, H3K27ac, H3K27me3 and H3K9me3, were used. 10 states were initially used and analogous states were subsequently pooled together resulting in five distinct states: active promoter (P), poised Promoter (Pp), active enhancers (E), repressed region (R) and background (B).

**Transcription factor binding site analysis.** TFBS analysis was performed using the Uniform TFBS (wgEncodeAwgTfbsUniform) dataset associated with LCLs[64].

**Reporting summary.** Further information on research design is available in the Nature Research Reporting Summary linked to this article.

## Data availability

The data that support this study are available from the corresponding authors upon reasonable request. The microarray and sequencing data generated for this study have been deposited at Gene Expression Omnibus (GEO) under the accession numbers: GSE192767, GSE193038, GSE193310, GSE193311, GSE193312, GSE193314 and GSE193315. Additional data used in this study are available under the GEO accession numbers: GSM733752, GSM758559 and GSE125924[64,65] as well as in the Uniform TFBS (wgEncodeAwgTfbsUniform) dataset associated with LCLs[64]. Genome assemblies: Homo sapiens (human) genome assembly NCBI36 (hg18) (https://www.ncbi.nlm.nih.gov/assembly/GCF_000001405.12/), Homo sapiens (human) genome assembly GRCh37 (hg19) (https://www.ncbi.nlm.nih.gov/assembly/GCF_000001405.13/). Source data are provided with this paper.

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

## Acknowledgements

We are very grateful to the ATR-X families and normal controls who donated samples that made this work possible. We are grateful to A. Pellagatti, S. Roy and V. Steeples for their help with the microarray assays, to the Klose group (Department of Biochemistry, University of Oxford) for providing the material to perform the Bio-CAP assay, to the Fulga group for the production of S2 cells and to the Milne group for the crosslinked S2 cells and their valuable advice. We also thank C. Fisher for the HbH inclusion assay and his advice, to S. Butler for her precious help with cell culture, to B. Xella and D. Clynes for their help and advice, to R. Beagrie for his help with the erythroblast experiments and T. Rostron (WIMM sequencing core) provided sequencing services. The single-cell experiments were performed in the WIMM single-cell facility with the help of N. Ashley. The Computational Biology Research Group, WIMM helped with the bioinformatics. The cell sorting and analysis were performed in the WIMM FACS facility. This work was supported by the Medical Research Council (UK) [grant number MC_UU_12025/ unit programmes MC_UU_12009/3 (RJG) and MC_UU_12009/4 (DRH)] and Congenital Anaemia Network [UK charity no. 1176864](CS).

## Author contributions

J.T., R.J.G. and D.R.H. conceptualised the study and conceived the experimental design; J.T. generated most of the data and analyses. D.J.D. helped to carry out experiments and analyses and generated the RNA-seq and Capture-C data from erythroblasts. C.S. and P.L.C. performed the CD34+ HSPC purification. C.S. performed the CD34+ HSPC differentiation and FACS analyses. M.G. contributed to the bulk ATAC-seq on erythroblasts. J.M.B. carried out the RNA FISH. J.T and D.J.D. performed most of the bioinformatics analyses. J.M.T. provided bioinformatic pipelines and supervision. E.R.G. performed the t-SNE analysis of the scATAC-seq data. E.R. performed the power calculation and the analysis of the microarrays raw data. S.T. provided the pipeline for the repeatome analysis. R.S. provided the pipeline and guidance for the GenoSTAN analyses. J.R.H. provided supervision for the bioinformatic analysis and funding. J.M.T., E.R., J.R.H., D.R.H. and R.J.G. advised on the interpretation of the data. J.T., D.R.H. and R.J.G. wrote the original manuscript; all authors contributed to reviewing and editing the manuscript. D.R.H. and R.J.G. provided supervision and funding.

## Competing interests

The authors declare no competing interests.
