## [Peer Review File · Nature Communications]

The chromatin remodeller ATRX facilitates diverse nuclear processes, in a stochastic manner, in both heterochromatin and euchromatinREVIEWER COMMENTS

Reviewer #1 (Remarks to the Author):

Truch. J . et al. provide multi epigenomic findings regarding alterations in ATRX focusing on chromatin architecture and transcription. They demonstrate the efficiency of their ChIP-seq approach for ATRX, which is important for the field as some targets have been troublesome to perform. With their integrated data analysis, the authors demonstrate the association of ATRX with various regulatory elements in LCL, erythroblasts cells, and normal human donors, which is remarkable for the field. Some of the central findings provided here recapitulate earlier published studies in the field, as should be noted. Moreover, this large data could be further mined to strengthen the article. My comments are below.

1) Using ATAC-seq on LCL cells, the authors demonstrate that 87% of ATRX binding sites are associated with open chromatin regions, while they provide reference to approximately 9% of the human genome with open chromatin. It would be interesting to see what "exact percentage" of ATRX binding sites correspond to open chromatin, as reported in Song , LY, et al. Genome Research, 2011.

2) The authors determine the association of ATRX binding sites with annotated cis-regulated regions and have found seven chromatin states associated with these genomic loci. It would be interesting to further determine the various transcriptional events in these seven states, by highlighting various factors and building a model for gene-regulatory networks operating in them.

3) The authors performed ATRX ChIP on LCL, erythroblasts, and normal cells and highlight the association of ATRX with various regulatory regions. It would be interesting to see how the dynamics of ATRX is maintained in different cell types, apart from blood cells. For instance, the authors can perform the peak/gene distribution analysis for other cell types, where the published ChIP-seq data is available.

4) ATRX binding sites were found to be enriched with GATA motifs in erythroid cells and RUNX motifs in LCLs. This implies that the binding partner for ATRX could be cell type-specific. Confirmation of these cell-type specific binding interactions, either via IP-mass spec (global) or IP-western (specific factors), would be compelling. Also, what are the gene-sets enriched in these co-binding sites?

5) The work will be significantly enhanced if the authors can perform RNA-seq on at least a few cases, to better provide correlations between ATRX binding and differential gene expression.

6) The authors use 100kb windows surrounding ATRX binding sites to determine associations with differentially expressed genes. The authors should also consider ± 10 kb and ± 50 kb, as previous reports on ATRX ChIP-seq have used these criteria to define the association. Are there significant changes in the spectrum of associated genes?

7) To further strengthen the data, the authors should over-express ATRX in ATRX deficient cells/LCLs or establish a model and validate the expression status of HBA and HBM genes to provide more mechanistic insights on their regulation.

8) How does heterogeneity in ATRX expression correlate with HBB, HBA, and HBM expression on a single cell level?

9) Since the single-cell data is available, could they be examined more generally? What other pathways and molecular networks are significantly altered by ATRX loss?

Reviewer #2 (Remarks to the Author):

J. Truch and colleagues argue in their manuscript that they have proven a functional role of ATRX in facilitation of diverse nuclear processes, both in hetero- and eu-chromatin and "in a stochastic manner". To support their claims, they use a battery of different biochemical

assays. While I do appreciate the value of the measurements they have made, I am not completely convinced by their reasoning and analysis of these data.

The authors focus on the ATRX protein, which is a well known gene from a SWI-SNF family, whose mutations are associated with several diseases. Their starting point is an analysis of binding sites of the ATRX protein in the genome obtained with a specific, improved ChIP-Seq protocol. This in itself is a result that might be an interesting resource to many researchers in the field.

The authors analyze the binding patterns and focus on the promoter and non-methylated CpG-related binding sites in active genes. They postulate that not only is the ATRX protein co-localizing with these promoters in the genome, but it has instrumental role in the process of making chromatin accesible, Histone H3.3 deposition and trascription itself.

Then they proceed to analyze the specific locus of alpha-globin locus in several individuals with different ATRX deficiencies using scRNASeq and scATAC-Seq.

In my opinion, the data provided by this study can be very valuable to many researchers, however the m nuscript in its current form has several deficiencies in the way the reasoning is conducted. In particular:

1. In the first results section, the authors claim that their ChIP-Seq analysis both confirms Heterochromatin association of ATRX and a significant enrichment in euchromatin. It is based on the presence of vast minority of binding sites (113) associated with some heterochromatin mark (therefore confirming association with heterochromatin) and most peaks associated with active regions (as measured by ATAC-Seq). They compare the number of peaks associated with ATAC-Seq (87%) regions to the estimates of euchromatin in the genome (9%) to conclude that there is a significant enrichment. I find the reasoning in this part inconsistent. If indeed just the comparison of the proportion of peaks in ATAC-Seq regions "proves" association with euchromatin - it also proves statistical depletion in the rest of the genome. In my opinion this section needs to be better analyzed. In particular, there should be an analysis of the efficiency of their ChIP-Seq protocol in heterochromatin vs. open chromatin. As well as a more serious analysis of the binding pattern. From the results as they are presented (unless there is some serious issue with immunoprecipitation from heterochromatic regions) it would seem important to state clearly that the association with heterochromatin is a minor fraction of the occupancy, and the protein is mainly associated with transcribed regions.

2. In the second section, the authors claim that "these results confirm the specific interactions between ATRX and active promoters and enhancers and the associated chromatin modifications". I am not sure about the specificity claim here. I think that it needs to be addressed, that given what was established in the previous section (87% of peaks overlap with an ATAC-Seq region) we would expect enrichment both in enhancers and promoters. Fig2E supports that it seems to be a rather general correlation of profiles rather than a "specific interaction". A more detailed analysis would include also regions where there seem to be ATAC-Seq regions but no ATRX.

3. In another section, the authors claim that the "enrichment of ATRX [..]depends on the level of gene expression" based on the results shown in Figure 3j. I would consider it a strong wording. They clearly correlate with the expression level, however there is no way to prove dependence without more evidence.

4. The section dealing with the effects of ATRX mutation based on comparison between two cases of cells derived from patients suffering from ATR-X syndrome in comparison with

their healthy father is to some extent the most interesting (new, interesting data, potentially shedding light on the issue) and the most difficult to understand. The authors present the results of ATAC-Seq and H3K27ac ChIP-Seq analysis of the locus indicating mostly the same locations of the peaks, with quantitative differences in ChIP and ATAC-Seq enrichments between healthy donors and patients A and B. It is unclear to me if the measured difference is significant, especially that there are many issues with ChIP-Seq data being interpreted as a quantitative measure. Then comes the scATACseq analysis with the tSNE-clustering that is not entirely convincing (also the parts of the tSNE that the authors point to in Fig. 5 fgh , do not seem to correlate with the clusters determined by tSNA in fig 5e). Perhaps the most convincing parts are the figures 5jk, where the authors show that there is indeed breaking of the correlation between HBB and HBA/HBM relative expression levels. I not sure if the data is consistent with the author's hypothesis is consistent with the model that only the most differentiated cells are affected. To me the data shows overall (even in controls) that the most differentiated cells have the least correlated expression levels, _and_ that there seems to be some disruption of the linear trend between expression levels and differentiation state in cases.

Overall I find this study interesting and potentially impactful, however there are multiple cases where the reasoning seems to be overly simplified (as in the enrichment argument) or overly speculative. I would recommend publishing this study if the issues I pointed out are addressed.

Reviewer #3 (Remarks to the Author):

In this manuscript, Truch and colleagues investigate the role of ATRX in the chromatin. They show that ATRX is not only associated to heterochromatin, but also associated to active regulatory elements. The manuscript reports a large amount of chromatin and transcriptomics data from including cell lines (LCLs) and hematopoietic cells of healthy donors and patients with ATRX mutations. The analysis of the sequencing data is mostly well suitable, but improvements on visual and textual presentation are necessary. Also, some of the analyses are missing a statistical treatment. Moreover, analysis of the single cell data is rather simple, and some improvements might be of benefit for the manuscript.

Specific points:

1. Several important aspects of this manuscript are based on overlap of the ATRX peaks with distinct genomic features (fig. 1b, 1d, 1g, fig. 2a, 2b, 3a,3b,3f,3g,3h). However, none of these present any statistical assessment of these overlaps. Authors should consider the use of appropriate tests, as for example projection, Jaccard (see <https://doi.org/10.1371/journal.pcbi.1002529>) or Fischer Exact tests.

2. On a similar aspect authors report the association of ATRX binding to DEG genes (lines 277-280). What is the significant of this overlap? Is this more than expected by chance?

3. One of the peaks with loss of ATAC-seq, Runx3 binding and chromatin marks in PBX4 is indicated as an enhancer. However, there is also a small but clear H3K4me3 peak lost on cases. Authors should verify if this is not an alternative TSS or discuss this in the text.

4. The analysis of the scATAC/scRNA-seq is very simple and should be further improved.

4.1 First, it would be helpful in authors attempt to annotate the identity of the clusters. In page 19, authors indicate late differentiated erythroid cells are being affected. Are clusters 3

and 11 such cells? What about other discussed clusters (8 and 9)? Some plots showing the expression of markers of early vs. late erythropoiesis would support this interpretation.

4.2 How do authors measure, which clusters are enriched in case and control? A statistical evaluation will be also desired here.

4.3 There are several interesting analyses that could further be explored with scATAC-seq. For example, measuring the activity of TFs in the cells by using chromVar or CisTopic. Authors could check for TFs with activity specific to clusters 3 and 11 and validate the changes on Runx3 in a cluster specific manner.

4.4 Are ATRX mutations expected to be present in only some of the erythroblast cells? If this is the case, authors could verify sequences associated to distinct clusters to see if the mutations is associated to the heterogeneity of the HBB/HBA1/HBM genes.

4.5 Authors should also present a clustering/t-sne and cell type characterization for scRNA-seq in a similar way as for scATAC-seq.

Minor points:

Figure 1 – Which LCL cell line is shown in individual plots (1a and 1e). The text mentions 3 cell lines. Which one is used in these plots? The same apply for other panels showing ChIP-seq/chromatin profiles.

Figure 1 – Why is H3K9me3 not shown in panel 1e?

Figure 2 – Legend of panel a) could be improved. Please spell out P, Pp,

Figure 4 – The volcano plot (a) is very small and non-legible. Authors should increase their side and annotate relevant genes, i.e. down-regulated genes as NME4 discussed in the text.

Figure 5 – color code for panel a) is hard to follow. Authors should use a paired color code (two shades of red) for disease samples A and B.

Figure 5 - Panels 5f,5g and 5h could be replaced by a plot showing the distribution of the gene scores per cluster, i.e. using a violin plot for example.

Figure 5 – it is unclear what is exactly shown in panel i. Please improve the legend description.

Page 8 line 135 – Upon introduction of figure 1a, there is no reference or explanation of the BioCAP protocol.

Page 8 line 152- There is a missing reference.

Page 16 / Supplemental Figure S12 – Authors should mark relevant genomic regions (r1, r2, r3, ...) in genomic views reported in the supplement as done in the Fig. 5a.

Page 31/706 – Authors should provide a public link to any provide code. The one indicate in the text is possibly an internal file, which is not available for people outside the author’s institution.

Replies to reviewers

Reviewer #1 (Remarks to the Author):

Truch. J . et al. provide multi epigenomic findings regarding alterations in ATRX focusing on chromatin architecture and transcription. They demonstrate the efficiency of their ChIP-seq approach for ATRX, which is important for the field as some targets have been troublesome to perform. With their integrated data analysis, the authors demonstrate the association of ATRX with various regulatory elements in LCL, erythroblasts cells, and normal human donors, which is remarkable for the field. Some of the central findings provided here recapitulate earlier published studies in the field, as should be noted. Moreover, this large data could be further mined to strengthen the article. My comments are below.

1) Using ATAC-seq on LCL cells, the authors demonstrate that 87% of ATRX binding sites are associated with open chromatin regions, while they provide reference to approximately 9% of the human genome with open chromatin. It would be interesting to see what “exact percentage” of ATRX binding sites correspond to open chromatin, as reported in Song , LY, et al. Genome Research, 2011.

ATAC seq is currently the gold standard for open chromatin rather than the DNase1 hypersensitivity and FAIRE used in Song et al. so we have attempted to use ATAC-seq data to determine the “exact percentage” of ATRX binding sites correspond to open chromatin. Using ATAC-seq we find that 87% of ATRX binding sites are associated with open chromatin sites, whereas, ATRX binding sites are only present in 12% (7,355 out of 55,640 sites) of the open chromatin in LCL (Fig.1g). We have added a statistical analysis by comparing our results obtained at ATRX binding sites with a set of random fragments matching in size and number to the ATRX bound sites. In that way, we can directly evaluate the specificity of ATRX binding sites for each overlap in the study. We observe that only 6% of the random fragments overlap with ATAC-seq peaks (Supplemental Figure S3d). This statistical analysis is more relevant than referring to the data in Song et al to calculate open chromatin so we have removed this reference.

2) The authors determine the association of ATRX binding sites with annotated cis-regulated regions and have found seven chromatin states associated with these genomic loci. It would be interesting to further determine the various transcriptional events in these seven states, by highlighting various factors and building a model for gene-regulatory networks operating in them.

Establishing gene regulatory networks would be of interest but is not the focus of the work described here and could be the purpose of a new study. Nevertheless, based on the referee’s suggestions we performed (i) a GO analysis and (ii) an enrichment analysis of the transcription factors for each subgroup of ATRX binding sites based on the seven states of GenoSTAN annotation (see **Supplemental Figure S5**), thus shedding some light on the factors and pathways enriched in each subgroup.

3) The authors performed ATRX ChIP on LCL, erythroblasts, and normal cells ? and highlight the association of ATRX with various regulatory regions. It would be interesting to see how the dynamics of ATRX is maintained in different cell types, apart from blood cells. For instance, the authors can perform the peak/gene distribution analysis for other cell types, where the published ChIP-seq data is available.

This would be an interesting analysis if published data sets of sufficient quality from different cell types were available to allow accurate peak calling. Previous data from our laboratory and others is plagued by high background (as illustrated below by comparing ATRX ChIP-seq datasets for a particular region in erythroblasts from Law et al 2010 and the current manuscript). To obtain data that is comparable in a meaningful way, we would have to perform extensive ChIP studies, using the modified protocol used in this report, of other cell types which is beyond the scope of the current work.

Nevertheless our conclusions are supported by the work of Danussi et al. 2018. Although not technically comparable, in a different species and analysed in different cell type, the report from Danussi came to similar conclusions. In mouse neuroprogenitor cells, in the ATRX ChIP-seq dataset which displayed a low background, ATRX binding sites were mainly intragenic, overlapping with promoters and gene bodies. We refer to this report in the discussion section.

4) ATRX binding sites were found to be enriched with GATA motifs in erythroid cells and RUNX motifs in LCLs. This implies that the binding partner for ATRX could be cell type-specific. Confirmation of these cell-type specific binding interactions, either via IP-mass spec (global) or IP-western (specific factors), would be compelling. Also, what are the gene-sets enriched in these co-binding sites?

GATA1 and RUNX1 are master regulators of the erythroid and lymphoid transcriptional programmes respectively. Consequently, these motifs are greatly enriched at the cis-regulatory elements of the genes expressed in these cell types. By contrast ATRX as a chromatin remodelling factor is likely to be recruited as a result of activation of such regulatory elements and plays an accessory role. The fact that these factors are found together at active regulatory elements does not imply that they interact by direct protein-protein interactions.

5) The work will be significantly enhanced if the authors can perform RNA-seq on at least a few cases, to better provide correlations between ATRX binding and differential gene expression.

We decided to perform microarray on an extensive number of ATRX cases and controls to ensure we could examine examples showing robust changes in RNA expression. Microarrays are known to provide more accurate quantitative expression data. These expression data were validated by qPCR. RNA-seq, would undoubtedly confirm these examples and potentially provide some other examples. However, it would add little to the main aim of this work which is to understand the principles by which ATRX binds and affects gene expression.

6) The authors use 100kb windows surrounding ATRX binding sites to determine associations with differentially expressed genes. The authors should also consider ± 10 kb and ± 50 kb, as previous

reports on ATRX ChIP-seq have used these criteria to define the association. Are there significant changes in the spectrum of associated genes?

To address this point we have added the two suggested windows (10kb and 50kb) as well as inside the gene. We also checked if the LogFC could have any effect. All these results are now summarised in Table 1 and Supplemental Table 1 and They demonstrate the same GO terms and only minor changes in gene number (count).

7) To further strengthen the data, the authors should over-express ATRX in ATRX deficient cells/LCLs or establish a model and validate the expression status of HBA and HBM genes to provide more mechanistic insights on their regulation.

This would be an interesting experiment and we have tried to do it. There are two currently unsurmountable problems in performing such experiments. The first is that it is not possible to efficiently transfect LCL cells: we have tried many times and to our knowledge, there is no efficient protocol to do so. The second is that mutant CD34+ cells, derived from very rare patients with ATRX syndrome are an extremely limited resource and we had to prioritise these cells for the other analyses. We will of course aim to establish other models to address this in the future but it is beyond the scope of this study.

8) How does heterogeneity in ATRX expression correlate with HBB, HBA, and HBM expression on a single cell level?

We have analysed this and found no correlation between ATRX expression with either HBB, HBA or HBM expression on a single cell level as illustrated below in the scatter plots showing the expression data and in Supplemental Figure S16b.

9) Since the single-cell data is available, could they be examined more generally? What other pathways and molecular networks are significantly altered by ATRX loss?

We have looked at this but there is a caveat. In many single cell analyses there is a mixture of cells representing different lineages. By contrast, our study analyses only cells from the erythroid lineage and consequently these cells predominantly express erythroid genes. Non-erythroid genes are poorly represented in the sequences at the coverage we could reasonably obtain.

Reviewer #2 (Remarks to the Author):

J. Truch and colleagues argue in their manuscript that they have proven a functional role of ATRX in facilitation of diverse nuclear processes, both in hetero- and eu-chromatin and "in a stochastic manner". To support their claims, they use a battery of different biochemical assays. While I do appreciate the value of the measurements they have made, I am not completely convinced by their reasoning and analysis of these data.

The authors focus on the ATRX protein, which is a well known gene from a SWI-SNF family, whose mutations are associated with several diseases. Their starting point is an analysis of binding sites of the ATRX protein in the genome obtained with a specific, improved ChIP-Seq protocol. This in itself is a result that might be an interesting resource to many researchers in the field.

The authors analyze the binding patterns and focus on the promoter and non-methylated CpG-related binding sites in active genes. They postulate that not only is the ATRX protein co-localizing with these promoters in the genome, but it has instrumental role in the process of making chromatin accesible, Histone H3.3 deposition and trascripion itself.

Then they proceed to analyze the specific locus of alpha-globin locus in several individuals with different ATRX deficiencies using scRNASeq and scATAC-Seq.

In my opinion, the data provided by this study can by very valuable to many researchers, however the mnuscript in its current form has several deficiencies in the way the reasoning is conducted. In particular:

1. In the first results section, the authors claim that their ChIP-Seq analysis both confirms Heterochromatin association of ATRX and a significant enrichment in euchromatin. It is based on the presence of vast minority of binding sites (113) associated with some heterochromatin mark (therefore confirming association with heterochromatin) and most peaks associated with active regions (as measured by ATAC-Seq). They compare the number of peaks associated with ATAC-Seq (87%) regions to the estimates of euchromatin in the genome (9%) to conclude that there is a significant enrichment. I find the reasoning in this part inconsistent. If indeed just the comparison of the proportion of peaks in ATAC-Seq regions "proves" association with euchromatin - it also proves statistical depletion in the rest of the genome. In my opinion this section needs to be better analyzed. In particular, there should be an analysis of the efficiency of their ChIP-Seq protocol in heterochromatin vs. open chromatin. As well as a more serious analysis of the binding pattern. From the results as they are presented (unless there is some serious issue with immunoprecipitation from heterochromatic regions) it would seem important to state clearly that the association with heterochromatin is a minor fraction of the occupancy, and the protein is mainly associated with transcribed regions.

The binding of ATRX at heterochromatin has been established in many previous studies by a number of different methods including indirect immunofluorescence (McDowell et al 1999) and chromatin immunoprecipitation analysed by hybridisation (ChIP-chip, Law et al 2010). Short read technology as used here in ChIP-seq will, because of its repetitive nature and consequent problems of mapping, underestimate the binding at heterochromatin. Nevertheless it is still possible to demonstrate, if not quantify, binding at heterochromatin. We used two approaches in order to analyse the ATRX enrichment at such regions. First we used a direct mapping approach against the repeatome (Repeatome Analysis in the methods section) which confirmed the enrichment of ATRX for heterochromatic regions (Supplemental figure S2a). Secondly, we use a standard ChIP-seq analysis approach (analysis of raw sequencing data from ChIP-seq, ATAC-seq and Bio-CAP-seq experiments in the method section). In this case, the majority of heterochromatin regions because they are repetitive are discarded in the mapping. Hence using this method we found that only 10% of sequences map to repressed heterochromatic regions. Whereas the presence of ATRX in heterochromatin has been previously described, the distribution of ATRX in euchromatin has been uncharted. The major point of this paper is that using a much-improved ChIP assay, we have now defined the distribution of ATRX in euchromatin.

2. In the second section, the authors claim that "these results confirm the specific interactions between ATRX and active promoters and enhancers and the associated chromatin modifications". I am not sure about the specificity claim here. I think that it needs to be addressed, that given what was established in the previous section (87% of peaks overlap with an ATAC-Seq region) we would expect enrichment both in enhancers and promoters. Fig2E supports that it seems to be a rather general correlation of profiles rather than a "specific interaction". A more detailed analysis would include also regions where there seem to be ATAC-Seq regions but no ATRX.

To address this point, we have analysed regions where there are ATAC-seq peaks but no ATRX. To accurately compare these sites, we also performed the analysis on ATRX binding sites keeping only those overlapping with ATAC-seq (Supplemental Figure S5). Comparing regions with ATAC-seq peaks but no ATRX with either all the ATRX binding sites, or the ATRX binding sites associated with least one ATAC-seq peak, confirmed in both cases a significant enrichment of active promoters and enhancers in ATRX bound regions (p-value < 2.2e-16, odds ratio = 2.4 and p-value < 2.2e-16, odds ratio = 2.5, respectively - Fisher's Exact Test). We have added these statistics in the manuscript and Figure. Besides, we agree that the use of "specific" was misleading and we have removed it.

3. In another section, the authors claim that the "enrichment of ATRX [...] depends on the level of gene expression" based on the results shown in Figure 3j. I would consider it a strong wording. They clearly correlate with the expression level, however there is no way to prove dependence without more evidence.

Indeed, we agree that the term "depends" was not the most appropriate and we changed it for "correlates".

4. The section dealing with the effects of ATRX mutation based on comparison between two cases of cells derived from patients suffering from ATR-X syndrome in comparison with their healthy father is to some extent the most interesting (new, interesting data, potentially shedding light on the issue) and the most difficult to understand. The authors present the results of ATAC-Seq and H3K27ac ChIP-Seq analysis of the locus indicating mostly the same locations of the peaks, with quantitative differences in ChIP and ATAC-Seq enrichments between healthy donors and patients A and B. It is unclear to me if the measured difference is significant, especially that there are many issues with ChIP-Seq data being interpreted as a quantitative measure.

We agree that ATAC-seq is not quantitative. However, we found good agreement between separate experiments: one using a bulk population and a second using single cell analysis that was then integrated to produce a population view. These results were in very good agreement, and both showed consistent changes in ATAC signal. By contrast the H3K27ac ChIP-seq was performed as a ChIP-Rx and normalised based on the *Drosophila melanogaster* S2 cells “spiked in” so is quantitative. Again these showed changes consistent with the changes seen in ATAC peaks. Although this was included in the GEO submission, it was not mentioned in the legend of the figure 5 and this explanation has now been included.

Then comes the scATACseq analysis with the tSNE-clustering that is not entirely convincing (also the parts of the tSNE that the authors point to in Fig. 5 fgh , do not seem to correlate with the clusters determined by tSNA in fig 5e).

We understand that our section on the single cell analyses needed to be clarified. Our datasets represent a relatively homogenous population of erythroid cells in contrast to more conventional single cell analyses performed on heterogenous populations containing several cell types. We apologise that these points were unclear in the original manuscript and have tried to improve this explanation in the revised text.

Single cell analysis can be performed to identify homogenous subpopulations within a heterogenous population containing several cell types and cluster analysis is most commonly performed to achieve this. For example a heterogenous dataset such as the 10X publicly available peripheral blood mononuclear cells (pbmc) contains several different cell types and can be annotated based on the clustering (see picture below: cisTopic analysis on 10X publicly available pbmc).

In this current study, however, we are analysing a population of cells that are relatively homogenous in their differentiation (as shown by FACS and cytoSpin and confirmed by single cell analysis) to identify subtle changes in a subgroup. The standard cluster analysis does not apply in this case. Indeed our cells are clustering in one major cluster (cell population) and this supports the robustness of our protocol. Consequently, displaying the Louvain clustering was misleading as the clustering observed does not distinguish cell types but may instead depend on slight differences such as maturity levels or differences in their cell cycle phases. We have now removed it (applying instead

an approach used in Johansson et al., 2021 which also focussed on a homogenous population of cells).

The purpose of our experiment was to focus on the *HBB*, *HBA* and *HBM* genes and assess if there was any correlation with the phenotype observed in patients. Of note, the presence of HbH inclusions in alpha-thalassaemia patients is due to an excess of HBB expression compared to HBA expression and so we were looking for difference in expression and accessibility at these loci. The annotated clustering in the cisTopic analysis of our scATAC-seq data was not wholly determined by accessibility at *HBB*, *HBA* and *HBM* and hence the accessibility at these genes is not limited to particular clusters in Figures 5 f,g and h. However, due to the limitation of the analysis tools available we needed to use this clustering to highlight the cells in the ATR-X case and control at particular regions of the tSNE to show where there were clear differences in the accessibility (as judged from ATAC-seq) of these genes.

The analysis of our scRNA-seq data also resulted in cells clustering mainly in one major population which is negative for the early CD34 marker and positive for the intermediate GPA marker (**Supplemental Figure S16b-c**). Although this involves a different model, this result is similar to that described in Johansson et al., 2021 who also focussed on a homogenous population of cells. Further analysis of our data showed that the cells mainly clustered based on their cell cycle phase rather than cell identity (see below in the UMAP analysis).

As described in the Seurat vignettes, we also observed that some G1 cells remain distinct from actively proliferating cells. Looking in more details at the gene expression and differentiation markers, we observed that in addition to clustering based on the cell cycle, the cells also cluster based on subtle variations in their stages of differentiation. Indeed, the cells on the left of the main cluster tend to be less differentiated than those on the right of the main cluster. Together, these results suggest that the distinct minor cluster at the bottom right of the main cluster contain cells in G1 that are slightly more advanced in the erythroid differentiation ((Supplemental Figures S16b-d).

In conclusion, standard cluster analysis usually performed on heterogenous population containing various cell types is not ideally adapted to our analysis in either scATAC-seq or scRNA-seq data. We therefore applied a more appropriate approach as seen in Johansson et al., 2021. We have updated the section on single cell analyses for clarity.

Perhaps the most convincing parts are the figures 5jk, where the authors show that there is indeed breaking of the correlation between HBB and HBA/HBM relative expression levels. I not sure if the data is consistent with the author's hypothesis is consistent with the model that only the most differentiated cells are affected. To me the data shows overall (even in controls) that the most differentiated cells have the least correlated expression levels, _and_ that there seems to be some disruption of the linear trend between expression levels and differentiation state in cases.

Our hypothesis is not that only the most differentiated cells are affected but that a subset of the most differentiated cells are the most affected (or at least those with a phenotype that we can detect). As described in the discussion, we suspect that ATRX deficiency may affect a subset of cells via a cumulative effect. The more the locus is active or the more the cells goes through rounds of cell divisions and differentiation, the more likely the cells are to acquired impairments in their chromatin environment. That is why the most differentiated cells would display a more extreme phenotype, whereas the preliminary impairments occurring in a subset of less differentiated cells may not be readily detectable.

In panels 5 j and k we show that statistical differences between the control and case based on the R^2 . The most differentiated cells (which have the highest levels of HBB) maintain correlated ratios of HBA/HBB expression whereas this correlation is lost in the case with ATR-X syndrome. To investigate this further, we repeated the analysis on the more differentiated cells. These results showed that the R^2 was conserved in a control but it dramatically decreased in cells derived from the ATR-X case confirming the disruption of the linear trend between expression levels and differentiation state in an individual with ATRX deficiency and alpha thalassaemia.

To clarify our hypothesis we have updated a section of the discussion as below:

“Here we have shown that in cells with a deleterious mutation in ATRX, chromatin accessibility at the alpha globin cluster is reduced in only a subset of cells at a later stage of differentiation in which the globin loci are more transcriptionally active. In single cells from this sub-population in which chromatin accessibility of alpha-globin is low, the accessibility of the beta globin gene is relatively high. This same subset of cells has an even greater reduction in the expression of the alpha-like gene *HBM*. These observations do not exclude the possibility that underlying defects associated with ATRX deficiency are already present at earlier stages of differentiation although below the level of detection and accumulate with rounds of cell division and differentiation.”

the reasoning seems to be overly simplified (as in the enrichment argument) or overly speculative. I would recommend publishing this study if the issues I pointed out are addressed.

Reviewer #3 (Remarks to the Author):

In this manuscript, Truch and colleagues investigate the role of ATRX in the chromatin. They show that ATRX is not only associated to heterochromatin, but also associated to active regulatory elements. The manuscript reports a large amount of chromatin and transcriptomics data from including cell lines (LCLs) and hematopoietic cells of healthy donors and patients with ATRX mutations. The analysis of the sequencing data is mostly well suitable, but improvements on visual and textual presentation are necessary. Also, some of the analyses are missing a statistical treatment. Moreover, analysis of the single cell data is rather simple, and some improvements might be of benefit for the manuscript.

Specific points:

1. Several important aspects of this manuscript are based on overlap of the ATRX peaks with distinct genomic features (fig. 1b, 1d, 1g, fig. 2a, 2b, 3a,3b,3f,3g,3h). However, none of these present any statistical assessment of these overlaps. Authors should consider the use of appropriate tests, as for example projection, Jaccard (see <https://doi.org/10.1371/journal.pcbi.1002529>) or Fisher Exact tests.

We have added these statistical analyses comparing our data with random fragments (*Supplemental Figure S3*) and using a Fisher Exact tests as suggested. Please find below the detail of all these changes:

fig. 1b

“ATRX binding sites were significantly enriched at TSS compared to matching size random fragments (p-value < 2.2e-16, odds ratio = 23.3, Fisher's Exact Test).”

fig. 1d

“ATRX binding sites were enriched in annotated CpG islands which were present in 29% (compared to 2% in matching size random fragments - p-value < 2.2e-16, odds ratio = 18.9, Fisher's Exact Test) (Figure 1d).”

“Only 113 ATRX binding sites were fully methylated on both alleles (p-value < 2.2e-16, odds ratio = 0.1, Fisher's Exact Test, compared to matching size random fragments) (Figure 1d)”

fig. 1g

“Comparison with matching size random fragments, ATRX is significantly enriched at regions of open chromatin (p-value < 2.2e-16, odds ratio = 108.9, Fisher's Exact Test).”

fig. 2a

“Of these, we observed a significant enrichment of ATRX binding sites at active enhancers (41% vs 22% of all sites, p-value < 2.2e-16, odds ratio = 2.5, Fisher's Exact Test) and promoters (32% vs 16% of all sites, p-value < 2.2e-16, odds ratio = 2.4, Fisher's Exact Test)”

fig. 2b

“(p-value < 2.2e-16, odds ratio = 141, Fisher's Exact Test compared to matching size random fragments).”

fig. 3a,3b

“As observed in LCLs, ATRX binding sites were mainly intragenic with a significant enrichment at TSS compared to matching size random fragments (p-value < 2.2e-16, odds ratio = 57.8, Fisher's Exact Test) and overlapping with open chromatin in erythroblasts (p-value < 2.2e-16, odds ratio = 138.5, Fisher's Exact Test) (Figure 3a and 3b).”

fig. 3f,3g and 3h

“As expected, shared peaks were enriched for gene promoters, CpG islands, open chromatin and PQS (p-value < 2.2e-16, odds ratio = 73.9, p-value < 2.2e-16, odds ratio = 83.9, p-value < 2.2e-16, odds ratio = 50.7 and p-value < 2.2e-16, odds ratio = 7.2, respectively, Fisher's Exact Test, compared to matching size random fragments) (Figure 3f, 3g and 3h)”

“Almost 95% of ATRX binding sites found in both LCLs and erythroblasts showed similar chromatin accessibility in both cell types (Figure 3g). Interestingly, most of the open chromatin sites were conserved in both LCLs and erythroblasts contrasting with the matching size random fragments in which the majority of the open chromatin regions were cell type specific (p-value < 2.2e-16, odds ratio = 33.1, Fisher's Exact Test).”

2. On a similar aspect authors report the association of ATRX binding to DEG genes (lines 277-280). What is the significant of this overlap? Is this more than expected by chance?

This is a good question. Our initial aim was to describe the environment of the most up or downregulated genes among the DEGs overlapping with ATRX but our analysis did not comprehensively address this referee’s point. To address the question if there is any link between DEGs and ATRX binding site localisation, we must consider the complete set of DEGs and ATRX binding sites.

Considering the probes tested in the microarray dataset, we found 149 probes corresponding to 83 DEGs with an ATRX binding site(s) in the gene body. Thus, we performed a Fisher t test on this dataset comparing with the DEGs not bound by ATRX and the non-DEGs bound and not bound by ATRX. We did the same analysis considering the different windows (10kb, 50kb, 100kb), in which ATRX binding sites could be detected or not. The analysis confirmed that DEGs are more likely to contain ATRX binding sites no matter the size of the window. We did the statistical analysis on both the probes and the DEGs, both were significant.

Probe based	Intragenic	10 kb	50 kb	100 kb
p-value	7.871e-11	1.656e-12	3.62e-07	7.47e-05
Odd ratio	2.51	2.1	1.7	1.6

gene based	Intragenic	10 kb	50 kb	100 kb
p-value	3.454e-06	1.512e-05	0.009595	0.01836
Odd ratio	1.95	1.81	1.42	1.39

The statistical analysis using the probes gave lower p-values and higher odds ratios, nevertheless, we chose to use the statistical analysis performed on genes in the manuscript as it is more biologically relevant. The results are mentioned in Supplemental Figure S10 and corrections have been made in the text as detailed below. If we consider only the DEG with a $|\logFC| > 0.5$, we add the criteria of expression level skewing the analysis and the p values increase. Nevertheless, the top pathway enrichments (GO analysis) remain very similar if we consider only the DEGs with a $|\logFC| > 0.5$ or all the DEGs (Table 1 and supplemental table 1). In consequence, we modified this section to clarify the paragraph as follow:

“Expression data were cross-referenced with the ATRX ChIP-seq data to analyse the distribution of the ATRX binding sites in the environment of the DEGs. Considering the complete set of DEGs and ATRX binding sites, we found a total of 83 DEGs containing at least one intragenic ATRX binding site and a total of 98 DEGs (10 kb window), 128 DEGs (50 kb window) and 156 DEGs (100 kb window) with at least one ATRX binding sites within these windows (p-value = 3.454e-06, odds ratio = 1.95, p-value = 1.512e-05, odds ratio = 1.81, p-value = 0.009595, odds ratio = 1.41, p-value = 0.01836, odds ratio = 1.39, respectively, Fisher's Exact Test) that could include genes directly affected by ATRX mutations (Supplemental Figure S10c).”

3. One of the peaks with loss of ATAC-seq, Runx3 binding and chromatin marks in PBX4 is indicated as an enhancer. However, there is also a small but clear H3K4me3 peak lost on cases. Authors should verify if this is not an alternative TSS or discuss this in the text.

This site is not described as an alternative TSS in UCSC and the histone modifications are more in line with it being an enhancer as is the Capture-C in Suppl Fig S11. ChromHMM refers to this region as an enhancer and most enhancers are marked by H3K4me3 although to a much lower degree than H3K4me1.

4. The analysis of the scATAC/scRNA-seq is very simple and should be further improved.

4.1 First, it would be helpful in authors attempt to annotate the identity of the clusters. In page 19, authors indicate late differentiated erytroid cells are being affected. Are clusters 3 and 11 such cells? What about other discussed clusters (8 and 9)? Some plots showing the expression of markers of early vs. late erythropoiesis would support this interpretation.

We acknowledge that our section on the single cell analyses needed to be clarified as our datasets represent a homogenous population in contrast to more conventional single cell analyses done on heterogenous populations containing several cell types. We apologise that these points were unclear in the original manuscript and have tried to improve this explanation in the revised text. In addition to answering your question 4.1, the explanations below also include elements of answers for your question 4.3 and 4.5.

Single cell analysis can be performed to identify homogenous subpopulations within a heterogenous population containing several cell types and cluster analysis is most commonly performed to achieve this. For example a heterogenous dataset such as the 10X publicly available peripheral blood mononuclear cells (pbmc) contains several different cell types and can be annotated based on the clustering (see picture below: cisTopic analysis on 10X publicly available pbmc).

In this current study, however, we are analysing a population of cells that are very homogenous in their differentiation (as shown by FACS and cytopsin and confirmed by single cell analysis) and we are trying to identify subtle changes in a subgroup. The standard cluster analysis does not apply in this case. Indeed our cells are clustering in one major cluster (cell population) and this supports the robustness of our protocol. Consequently, displaying the Louvain clustering was misleading as the clustering observed does not distinguish cell types but may instead depend on slight differences such as maturity levels or differences in their cell cycle phases. We have now removed it (applying instead an approach used in Johansson et al., 2021 which also focussed on a homogenous population of cells).

The purpose of our experiment was to focus on the *HBB*, *HBA* and *HBM* genes and assess if there was any correlation with the phenotype observed in patients. To note, the presence of HbH inclusions in alpha-thalassaemia patients is due to an excess of HBB expression compare to HBA expression and so we were looking for difference in expression and accessibility at these loci. The annotated clusters in the cisTopic analysis of our scATAC-seq data was not wholly determined by accessibility at *HBB*, *HBA* and *HBM* and hence the accessibility at these genes is not limited to particular clusters in Figures 5 f,g and h. However, due to the limitation of the analysis tools available we needed to use this clustering to highlight the cells in case and control at particular regions of the tSNE to show where there were clear differences in the accessibility of these genes.

The analysis of our scRNA-seq data also resulted in cells clustering mainly in one major population which is negative for the early CD34 marker and positive for the intermediate GPA marker (**Supplemental Figure S16b-c**). Although this involves a different model, this result is similar to that described in Johansson et al., 2021 who also focussed on a homogenous population of cells.

Further analysis of our data showed that the cells mainly clustered based on their cell cycle phase rather than cell identity (see below in the UMAP analysis).

As described in the Seurat vignettes, we also observed that some G1 cells remain distinct from actively proliferating cells. Looking in more details at the gene expression and differentiation markers, we observed that on top of the cell cycle clustering, the cells also cluster based on subtle variations in their differentiation states. Indeed, the cells on the left of the main cluster tend to be less differentiated than the one on the right of the main cluster. All together these results suggest that the distinct minor cluster at the bottom right of the main cluster contain cells in G1 that are slightly more advanced in the erythroid differentiation ((Supplemental Figures S16b-d). In conclusion, standard cluster analysis usually performed on heterogenous population containing various cell types is not ideally adapted to our analysis in either scATAC-seq or scRNA-seq data. We therefore applied a more appropriate approach as seen in Johansson et al., 2021. We have updated the section on single cell analyses for clarity.

We originally defined the more differentiated cells based on their level of HBB expression as the differentiation of CD34+ progenitors results in an increase in HBB expression. We have now added additional markers of differentiation, further supporting our hypothesis (Supplemental Figure S16). During differentiation, CD34 expression is lost across differentiation and cells are negative for this marker in scRNA-seq and cells become positive for GYPA (also known as GPA or CD235) as we can also observe in the scRNA-seq. Using a min.cutoff = "q5" (Vector of minimum cutoff values for each feature = quartile 5) we can visualise the cells with a high expression of GPA. Finally, *SLC4A1* (band 3 or CD233) is a marker of late stage of differentiation. In both scATAC and scRNAseq, high signal in HBB tends to correlate with higher signal of GPA and *SLC4A1*.

4.2 How do authors measure, which clusters are enriched in case and control? A statistical evaluation will be also desired here.

We originally concluded that clusters enriched in cases over controls based on the cell numbers in the clusters normalising for the total number of cells in ATR-X cases and controls. We have now added a statistical evaluation supporting our observation.

In the manuscript:

“This identified a subpopulation of cells, predominantly comprised of cells from ATR-X cases (p-value < 2.2e-16, odds ratio = 1.6, Fisher's Exact Test) (most of which belong to the ATR-X case 1 with alpha-thalassaemia (p-value < 2.2e-16, odds ratio = 2.3, Fisher's Exact Test)), in which despite a high chromatin accessibility at the *HBB* locus we found low accessibility at the *HBA* and *HBM* loci (encircled with a red line in Figure 5f-i).”

4.3 There are several interesting analyses that could further be explored with scATAC-seq. For example, measuring the activity of TFs in the cells by using chromVar or CisTopic. Authors could check for TFs with activity specific to clusters 3 and 11 and validate the changes on Runx3 in a cluster specific manner.

As explained in response to 4.1 and in the methods section of the manuscript, we used cisTopic to further explore the scATAC-seq dataset. Moreover, Runx3 is a lymphoid (not erythroid) transcription factor. Regarding GATA1 (an erythroid transcription factor) our data suggest that ATRX deficiency could affect a subset of GATA1 binding sites in erythroblasts (for example at the R1 enhancer) rather than changing GATA1 expression or chromatin accessibility at the GATA1 locus. Nevertheless we have added the GATA1 and ATRX signals in both scATAC-seq and scRNA-seq (Supplemental Figure S16).

4.4 Are ATRX mutations expected to be present in only some of the erythroblast cells? If this is the case, authors could verify sequences associated to distinct clusters to see if the mutations is associated to the heterogeneity of the HBB/HBA1/HBM genes.

No, the ATR-X cases have a germline mutation which is present in all cells.

4.5 Authors should also present a clustering/t-sne and cell type characterization for scRNA-seq in a similar way as for scATAC-seq.

As shown above, the single cell analysis undertaken here is not designed to cluster cells on cell type. There are subtle differences between ATR-X cases and controls shown in a cluster analysis but we found the best way to highlight the correlations and differences was via the scatter plots in Fig5i and j. We, nevertheless, have added the t-sne analysis for the scRNA-seq including the signal for the markers of differentiation, ATRX and the GATA1 transcription factor in addition to HBB, HBA and HBM (Supplemental Figure S16). On the UMAP analysis we can visualise a major population. Furthermore, the cells in both patient and control mainly clustered based on their cell cycle phase and slight differences in their maturation stages (Supplemental Figure S16b-d). A reduced HBA/HBM signal in a subpopulation of patient's cells with a high HBB signal can also be observed (highlighted by the red arrow in Supplemental Figure S16e).

Minor points:

Figure 1 – Which LCL cell line is shown in individual plots (1a and 1e). The text mentions 3 cell lines. Which one is used in these plots? The same apply for other panels showing ChIP-seq/chromatin profiles.

All three cell lines have been used for these plots. We have now added the n values and more information are also available in Supplemental Table S2 and on the GEO database that we have submitted as described in the data availability section.

Figure 1 – Why is H3K9me3 not shown in panel 1e?

H3K9me3 has now been added.

Figure 2 – Legend of panel a) could be improved. Please spell out P, Pp,

The legend in the figure 2 have been remade.

Figure 4 – The volcano plot (a) is very small and non-legible. Authors should increase their size and annotate relevant genes, i.e. down-regulated genes as NME4 discussed in the text.

The volcano plot has been remade and a subset of down regulated genes has been highlighted including NME4.

Figure 5 – color code for panel a) is hard to follow. Authors should use a paired color code (two shades of red) for disease samples A and B.

The picture has been remade with the updated color code (two shades of red) for disease samples from ATR-X case 1 and 2. (To note, samples A and B have been renamed case1 and case2 and don2 and don3 have been renamed Ctr2 and Ctr3 for ease of identification in the erythroblast data submitted to GEO).

Figure 5 - Panels 5f,5g and 5h could be replaced by a plot showing the distribution of the gene scores per cluster, i.e. using a violin plot for example.

As explained in our reply to question 4, plots per clusters are not the most appropriate for our data. We concluded that the best way to present these data was to use box plots highlighting the first quartile, median (highlighted by an arrow) and third quartiles as well as the outliers instead of violin plots that are not as straightforward to interpret (picture below).

Figure 5 – it is unclear what is exactly shown in panel i. Please improve the legend description.

We have updated the legend description in now panel h (corresponding previously to panel i).

Page 8 line 135 – Upon introduction of figure 1a, there is no reference or explanation of the BioCAP protocol.

We have now included this information in the figure legend.

Page 8 line 152- There is a missing reference.

This was a technical error in formatting the manuscript and should refer to Figure 1f rather than a reference.

Page 16 / Supplemental Figure S12 – Authors should mark relevant genomic regions (r1, r2, r3, ...) in genomic views reported in the supplement as done in the Fig. 5a.

The regions have now been marked.

Page 31/706 – Authors should provide a public link to any provide code. The one indicate in the text is possibly an internal file, which is not available for people outside the author's institution.

We have now corrected this section by referring to the relevant manuscript (which include a github access for source code: <https://github.com/Hughes-Genome-Group/NGseqBasic/releases>).

REVIEWERS' COMMENTS

Reviewer #1 (Remarks to the Author):

I appreciate the efforts the authors have made in this revision. I have no further issues with this manuscript.

Reviewer #3 (Remarks to the Author):

I am pleased to see that the authors considered the comments in particular the inclusion of statistics for genome interval analysis and clarifications regarding the analysis of the single cell data. The computational and statistical analysis of the manuscript are appropriate and do support the manuscript results.